# Generalized Schrödinger Bridge on Graphs

**Panagiotis Theodoropoulos** [1]  **Juno Nam** [2]  **Evangelos Theodorou** [1][†]  **Jaemoo Choi** [1][†]

## Abstract

Transportation on graphs is a fundamental challenge across many domains, where decisions must respect topological and operational constraints. Despite the need for actionable policies, existing graph-transport methods lack this expressivity. They rely on restrictive assumptions, fail to generalize across sparse topologies, and scale poorly with graph size and time horizon. To address these issues, we introduce Generalized Schrödinger Bridge on Graphs (GSBoG), a novel scalable data-driven framework for learning executable controlled continuous-time Markov chain (CTMC) policies on arbitrary graphs under state cost augmented dynamics. Notably, GSBoG learns trajectory-level policies, avoiding dense global solvers and thereby enhancing scalability. This is achieved via a likelihood optimization approach, satisfying the endpoint marginals, while simultaneously optimizing intermediate behavior under state-dependent running costs. Extensive experimentation on challenging real-world graph topologies shows that GSBoG reliably learns accurate, topology-respecting policies while optimizing application-specific intermediate state costs, highlighting its broad applicability and paving new avenues for cost-aware dynamical transport on general graphs.

## 1. Introduction

Graphs are a ubiquitous abstraction for complex systems (Easley & Kleinberg, 2010), providing a common representation via networks for the state space of diverse domains such as supply–demand systems (Zhang et al., 2003), road and traffic infrastructure (Chiu et al., 2011b), communication and social networks (Newman, 2003), power

[1]Georgia Institute of Technology [2]Massachusetts Institute of Technology. Correspondence to: Jaemoo Choi <jchoi843@gatech.edu>, Evangelos Theodorou <evangelos.theodorou@gatech.edu>.

*Proceedings of the 43rd International Conference on Machine Learning*, Seoul, South Korea. PMLR 306, 2026. Copyright 2026 by the author(s).

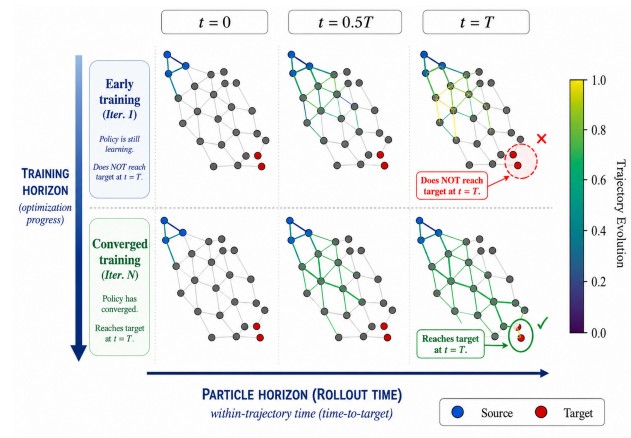

*Figure 1.* Learning graph routing policies from source (blue) to target (red) nodes, across time $t$ and training iterations. Edge intensity and color follows the trajectory-evolution colormap, illustrating how the learned policy progressively reallocates flow from the source region toward the target, as training converges.

grids (Pagani & Aiello, 2013), and even discrete models of molecular kinetics (Husic & Pande, 2018). In these settings, nodes represent discrete states, such as entities, objects, or data points, while edges encode feasible transitions between states, together with associated costs, often subject to physical constraints including capacity limits, congestion, and throughput (Bertsekas, 1998).

Graph transportation problems are naturally posed as probability distributions matching problems over nodes (e.g., supplies and demands). Therefore, optimal transport (OT) on graphs is typically used to offer a principled method of transporting the mass (Peyré & Cuturi, 2017). These frameworks usually output a single static coupling that matches the source and target distributions (Essid & Solomon, 2018; Peyré & Cuturi, 2017). While this coupling specifies how much mass should move between nodes in aggregate, it does not describe how mass should move over time, nor does it yield an executable control policy on the graph (Chizat et al., 2018). Consequently, intermediate trajectories are not optimized or controlled at deployment (O'Connor et al., 2022).

In real deployments, transportation is inherently time-dependent; hence, it is important to specify how mass moves through the network over time, not just a static plan (Chiu et al., 2011a). The core challenge is to construct a feasible, topology-respecting evolution that controls intermedi-

*Table 1.* Comparison of the Generalized Schrödinger Bridge problem in continuous space $\mathbb{R}^d$, and arbitrary discrete space $\mathcal{X}$.

| | Continuous state space $\mathbb{R}^d$ | Discrete State Space $\mathcal{X}$ |
|---|---|---|
| Reference Dyn. $p^r$ | $dX_t = r_t(X_t)\,dt + \sigma_t\,dW_t,\ X_0 \sim \mu$ | CTMC $(X_t)$ with transition rate $(r_t)$, $X_0 \sim \mu$ |
| Controlled Dyn. $p^u$ | $dX_t = (r_t + \sigma_t u_t)\,dt + \sigma_t\,dW_t,\ X_0 \sim \mu$ | CTMC $(X_t)$ with transition rate $(u_t)$, $X_0 \sim \mu$ |
| GSB | $\mathbb{E}_{X \sim p^u}\left[\int_0^1 f_t(X_t, p_t^u)\,dt\right] + \mathrm{KL}(p^u \,\|\, p^r)$ | |
| Continuity Equation | $\partial_t p_t^u = -\nabla \cdot \left((r_t + \sigma_t u_t)\,p_t^u\right) + \frac{\sigma_t^2}{2}\Delta p_t^u$ | $\partial_t p_t^u(x) = \sum_{z \in \mathcal{X}} u_t(x, z)\,p_t^u(z)$ |
| Iterative Proportional Fitting | Eqs. (10) and (11) | Eqs. (26) and (27) |
| Temporal Difference | Eq. (16) (Liu et al., 2022b) | Eqs. (28) and (29) |

ate distributions while minimizing network costs and accommodating operational constraints such as capacities and time variation (Peyré & Cuturi, 2017; Koch & Nasrabadi, 2014). This motivates dynamical formulations that optimize time-indexed flows and explicitly model distributional evolution on network domains (Burger et al., 2023). In this vein, Chow et al. (2022) introduced a dynamic Schrödinger bridge method to construct stochastic, time-indexed flows on graphs by solving Hamiltonian flows in the probability simplex over all nodes. However, in practice, existing dynamical approaches can be computationally expensive at scale and fragile on large, sparse networks, often needing to trade accuracy for tractability, implementing approximate schemes (Arqué et al., 2022; Facca et al., 2024).

In this work, we introduce **Generalized Schrödinger Bridge on Graphs** (GSBoG), an iterative and scalable framework for learning finite-horizon transport policies on general graphs. Closest to our setting is the dynamical SB on graphs (Chow et al., 2022); however, rather than solving global time expanded flows over the entire graph, our GSBoG adopts a particle-based and data-driven formulation, which alleviates the scalability challenges of existing graph-based approaches. Specifically, we extend generalized Schrödinger bridge (GSB) principles (Liu et al., 2022a; Chen et al., 2021; De Bortoli et al., 2021; Vargas et al., 2021) from continuous state spaces to graph-structured domains. Table 1 compares side-by-side the GSB between the continuous $\mathbb{R}^d$ and discrete state $\mathcal{X}$ space. Analogous to the continuous-state setting, GSBoG supports general running costs depending on both state and distribution, arbitrary source–target transport, and general graph topologies. To the best of our knowledge, this work is the first to formulate GSB transport on graphs in a data-driven manner. This setting is fundamentally different from recent discrete diffusion or SB approaches (Kim et al., 2024; Ksenofontov & Korotin, 2025; Lou et al., 2024), which primarily focus on graph generation or discrete-state synthesis. In contrast, GSBoG addresses distributional transport from a given source to a target distribution on a fixed graph, as illustrated in Figure 1, while optimizing trajectory-level behavior while also accounting for application-specific cost functionals.

Precisely, we formulate a GSB problem on graphs based on continuous-time Markov chain (CTMC) dynamics. We derive discrete optimality conditions that parallel the continuous-state formulation, providing a principled graph-based analogue of GSBs. Building on this analysis, we develop a data-driven iterative proportional fitting algorithm with an associated temporal-difference objective, enabling transport between arbitrary distributions on graphs while minimizing general cost functionals. Finally, we demonstrate the broad applicability of our proposed framework through representative examples on challenging real-world graphs, enabling cost-aware dynamical transport on general networked systems. Our contributions are summarized as follows:

- We introduce GSBoG, a first data-driven generalized Schrödinger bridge on *arbitrary graphs*, achieving simultaneously endpoint marginal distribution matching and optimization of the intermediate trajectory under general running costs.

- We formulate graph transport as controlled particle motion under CTMC dynamics and derive tractable objectives via path-space characterization.

- Extensive experimentation that GSBoG learns topology-respecting transport policies with accurate endpoint matching and improved intermediate-cost behavior compared to baseline methods.

## 2. Generalized Schrödinger Bridge Problem

**Notations.** Let $\mathcal{X}$ be a continuous state space (assumed to be a Polish space), and let $\mu, \nu \in \mathcal{P}(\mathcal{X})$ denote the source and target probability measures on $\mathcal{X}$, respectively. Here, $\mathcal{P}(\mathcal{X})$ denotes the set of probability measures on $\mathcal{X}$. We consider a reference stochastic dynamics given by the stochastic differential equation (SDE)

$$dX_t = r_t(X_t)\,dt + \sigma_t\,dW_t, \quad X_0 \sim \mu, \qquad (1)$$

where $r : [0, 1] \times \mathcal{X} \to \mathcal{X}$ is a prescribed *base drift*, $\sigma : [0, 1] \to \mathbb{R}_{>0}$ is a time-dependent noise schedule, and $\{W_t\}_{t \in [0,1]}$ is a standard Brownian motion.

**Generalized Schrödinger Bridge (GSB) Problem**  The problem we consider in this work is a *generalized* setting of Schrödinger Bridge problem (GSB; Chen et al. (2014)). The transportation problem seeks a control function $u : [0,1] \times \mathcal{X} \to \mathcal{X}$ such that the controlled dynamics

$$\mathrm{d}X_t = \big(r_t(X_t) + \sigma_t u_t(X_t)\big)\,\mathrm{d}t + \sigma_t\,\mathrm{d}W_t, \ \ X_0 \sim \mu, \quad (2)$$

transports the source distribution $\mu$ at $t = 0$ to the target distribution $\nu$ at $t = 1$, i.e., $X_1 \sim \nu$, while the dynamics are influenced by a state- and marginal-dependent running cost. Let $f : [0,1] \times \mathcal{X} \times \mathcal{P}(\mathcal{X}) \to \mathbb{R}$ denote a cost function that may depend on the time, the state, and the time-$t$ marginal distribution. The GSB seeks the controlled probability path $p^u$ that solves

$$\min_{p^u} \ \mathbb{E}_{X \sim p^u}\left[\int_0^1 f_t(X_t, p_t^u)\,\mathrm{d}t\right] + \mathrm{KL}(p^u \,\|\, p^r), \quad (3)$$

$$\text{s.t. } \partial_t p_t^u = -\nabla\cdot\big((r_t + \sigma_t u_t)\,p_t^u\big) + \frac{\sigma_t^2}{2}\,\Delta p_t^u, \quad (4)$$

$$p_0^u = \mu, \quad p_1^u = \nu,$$

where the PDE in Eq. (4) is the Fokker–Planck Equation (FPE) describing the time evolution of the controlled path marginal density $p_t^u$. Note that the Kullback–Leibler (KL) divergence $\mathrm{KL}(p^u\|p^r)$ regularizes the controlled dynamics toward the reference process, yielding an entropic optimal transport problem over path space. By applying Girsanov's theorem (Särkkä & Solin, 2019), the problem admits an equivalent control-theoretic formulation:

$$\min_u \ \mathbb{E}_{X \sim p^u}\left[\int_0^1 \tfrac{1}{2}\|u_t\|^2 + f_t\,\mathrm{d}t\right], \text{ s.t. } (4),\ X_0 \sim \mu,\ X_1 \sim \nu.$$

This formulation highlights the trade-off between control effort and task-dependent costs encoded by $f_t$.

**Hopf–Cole transform and generalized Schrödinger system**  From the SB theory, the optimal drift is given by $u_t = \sigma_t \nabla \log \varphi_t(X_t)$, where the functions $(\varphi, \widehat{\varphi})$ are referred as the Schrödinger potentials and satisfy the following set of coupled PDEs

$$\partial_t \varphi_t = -\nabla\varphi_t^\intercal r_t - \frac{\sigma_t^2}{2}\Delta\varphi_t + f_t\varphi_t, \quad \varphi_0\widehat{\varphi}_0 = \mu,$$
$$\partial_t \widehat{\varphi}_t = -\nabla\cdot(\widehat{\varphi}_t r_t) + \frac{\sigma_t^2}{2}\Delta\widehat{\varphi}_t - f_t\widehat{\varphi}_t, \quad \varphi_1\widehat{\varphi}_1 = \nu. \quad (5)$$

Thus, learning $(\nabla\log\varphi, \nabla\log\widehat{\varphi})$ suffices to solve the (generalized) SB problem.

**FBSDE Representation**  Rather than directly solving the coupled PDE system Eq. (5), we can obtain local solutions along high-probability paths through forward–backward stochastic differential equation (FBSDE) representation via Itô's formula (Liu et al., 2022b). For convenience, let us define along the forward trajectories $Y_t := \log\varphi_t, \hat{Y}_t :=$ $\log\hat{\varphi}_t$, and their gradients $Z_t := \sigma_t\nabla\log\varphi_t,\ \hat{Z}_t := \sigma_t\nabla\log\hat{\varphi}_t$. Then a representative form of the corresponding FBSDE is given by

$$dX_t = \big(r_t(X_t) + \sigma_t Z_t\big)\,dt + \sigma_t dW_t, \quad X_0 \sim \mu, \quad (6)$$

$$dY_t = \Big(\tfrac{1}{2}\|Z_t\|^2 + f_t\Big)\,dt + Z_t^\intercal dW_t, \quad (7)$$

$$d\widehat{Y}_t = \Big(\tfrac{1}{2}\|\widehat{Z}_t\|^2 + \nabla\cdot(\sigma\widehat{Z}_t - r_t)$$
$$+ \widehat{Z}_t^\intercal Z_t - f_t\Big)\,dt + \widehat{Z}_t^\intercal dW_t, \quad (8)$$

Equivalently, we can obtain a similar triplet of FBSDEs for the time reversed dynamics $s := 1 - t$, in which case $d\tilde{X}_s$ is controlled by the $\widehat{Z}$ (see Liu et al. (2022b)). Note that these expressions hold under the assumption that the base drift $b_t(x)$ is linear in the state variable $x$.

**gIPF Objective**  The coupled FBSDEs in Eqs. (6) and (8) provide a stochastic representation of the PDEs in Eq. (5). Consequently, rather than solving the PDEs in the entire function space, it suffices to solve them locally around high probability regions characterized by the FBSDEs. In particular, notice that the endpoint likelihood can be cast as follows

$$\log\mu(X_0) = \mathbb{E}_{p_{1|0}^Z}[\log\nu(X_1)] - \mathbb{E}_{p_{\cdot|0}^Z}\left[\int_0^1 dY_t + d\widehat{Y}_t\right], \quad (9)$$

Thus, to maximize the likelihood, we should minimize the terminal term. Hence, expanding Eqs. (7) and (8), we obtain the gIPF objective updating the backward policy $\widehat{Z}_t$:

$$\mathcal{L}_{\mathrm{IPF}}^Z(\widehat{Z}) := \mathbb{E}_\mu\mathbb{E}_{p_{\cdot|0}^Z}\left[\int_0^1 dY_t + d\widehat{Y}_t\right] \quad (10)$$

$$= \mathbb{E}_{p^z}\left[\int_0^1 \tfrac{1}{2}\|\widehat{Z}_t\|^2 + \widehat{Z}_t^\intercal Z_t + \nabla\cdot(\sigma\widehat{Z}_t - r_t)dt\right].$$

Similarly, the gIPF loss for updating $Z_t$ is:

$$\mathcal{L}_{\mathrm{IPF}}^{\widehat{Z}}(Z) := \mathbb{E}_\nu\mathbb{E}_{p_{\cdot|1}^{\widehat{Z}}}\left[\int_0^1 dY_t + d\widehat{Y}_t\right] \quad (11)$$

$$= \mathbb{E}_{p^{\hat{z}}}\left[\int_0^1 \tfrac{1}{2}\|Z_t\|^2 + Z_t^\intercal \widehat{Z}_t + \nabla\cdot(\sigma Z_t - r_t)dt\right].$$

Alternating minimization of Eqs. (10) and (11) can be viewed as successive KL projections between the corresponding paths, converging to the Schrödinger bridge (Vargas, 2021; De Bortoli et al., 2021).

**Temporal-Difference Objective**  However, in the generalized setting notice that the additional state cost terms in Eqs. (10) and (11) cancel out in the naive IPF aggregation. As a result, the IPF loss alone does not explicitly enforce consistency with the state- and distribution-dependent cost. To explicitly inject the running cost into learning, the potential dynamics are discretized, and via a temporal difference

loss (Liu et al., 2022b), deviations between adjacent increments are penalized forcing the learned potentials to be locally consistent with the cost-augmented dynamics. In practice, the gIPF algorithm alternates between minimizing the IPF objective and the TD objective, yielding an iterative scheme that enforces both marginal constraints and consistency with respect to the running cost.

## 3. Methodology

In this section, we introduce **Generalized Schrödinger Bridge on Graphs** (GSBoG), a framework for optimal transport on a given directed graph $\mathcal{G} = (\mathcal{X}, \mathcal{E})$. Here, $\mathcal{X} := \{1, \ldots, n\}$ denotes a finite set of nodes, and $\mathcal{E} \subset \mathcal{X} \times \mathcal{X}$ denotes a set of directed edges, where $(x, y) \in \mathcal{E}$ represents a directed transition from node $x$ to node $y$. Given a source distribution $\mu$ and a target distribution $\nu$ over the node set $\mathcal{X}$, our goal is to construct a controlled stochastic process on $\mathcal{G}$ that transports $\mu$ to $\nu$ while minimizing a generalized cost functional. The proposed framework can be viewed as a discrete, graph-structured counterpart of generalized Iterative Proportional Fitting (gIPF) (Liu et al., 2022a), with general state- and distribution-dependent cost functionals extended to continuous-time Markov chains on graphs.

We emphasize that GSBoG addresses a fundamentally different problem from prior data-driven works, such as DDSBM (Kim et al., 2024), which focus on generative modeling of graph structures. In contrast, our setting assumes a fixed graph topology and studies the problem of transporting probability mass over the given node space $\mathcal{X}$. To the best of our knowledge, this perspective has not been systematically investigated in the machine learning literature.

### 3.1. GSB Problem on Graphs

**Continuous-Time Markov Chains (CTMCs)** Let $(X_t)_{t \in [0,1]}$ be a stochastic process taking values in a finite state space $\mathcal{X}$. The process is characterized by its **transition rate** matrix $r = (r_t(y, x))_{x,y \in \mathcal{X}, \, t \in [0,1]}$, defined by

$$r_t(y, x) := \lim_{h \to 0} \frac{\Pr(X_{t+h} = y \mid X_t = x) - \mathbf{1}_{\{y=x\}}}{h},$$

where $\mathbf{1}_{\{\cdot\}}$ denotes the indicator function. Intuitively, $r_t(y, x)$ specifies the instantaneous rate of transition from state $x$ to state $y$ at time $t$.

We consider CTMCs whose transitions are constrained by a directed graph $\mathcal{G} = (\mathcal{X}, \mathcal{E})$. Accordingly, the transition rate $r_t$ is required to satisfy

$$\begin{cases} r_t(y, x) \geq 0, & \text{whenever } (x, y) \in \mathcal{X} \times \mathcal{X}, \\ r_t(y, x) = 0, & \text{whenever } (x, y) \notin \mathcal{E}, \\ r_t(x, x) = -\sum_{y \neq x} r_t(y, x). \end{cases} \quad (12)$$

for all $x, y \in \mathcal{X}$ and $t \in [0, 1]$. The second condition enforces the graph structure by prohibiting transitions along non-existent edges, while the last condition ensures mass conservation, i.e., $\sum_y r_t(y, x) = 0$ for all $x$.

**Generalized Schrödinger Bridge on Graphs** We consider the discrete analogue of Eq. (3) on $\mathcal{G}$. Let $u_t$ denote a controlled transition rate satisfying the same structural constraints. We denote by $p^r$ and $p^u$; the path measures induced by the base rate $r_t$ and the controlled rate $u_t$, respectively. In the following, we formulate the GSB problem on graphs with respect to the controlled transition rate $u_t$, by explicitly expressing the KL divergence between path measures of CTMC, the resulting optimization problem is written as

$$\min_u \ \underbrace{\mathbb{E}_{p^u} \mathbb{E}_t \left[ \sum_{y \neq X_t} \left( u_t \log \frac{u_t}{r_t} - u_t + r_t \right)(y, X_t) \right]}_{=: \mathrm{KL}(p^u \| p^r)} \quad (13)$$

$$+ \mathbb{E}_{p^u} \mathbb{E}_t \left[ f_t(X_t, p_t) \right] \quad \text{s.t.} \quad X_1 \sim \nu,$$

$$\partial_t p_t^u(x) = \sum_{y \in \mathcal{X}} u_t(x, y) p_t^u(y), \quad X_0 \sim \mu, \quad (14)$$

where Eq. (14) is the Continuity Equation (CE) that describes the time evolution of the controlled path $p_t^u$ on $\mathcal{G}$, serving as the discrete analogue of the FPE in Eq. (4). Note that the continuous-state dynamics constraint Eq. (4) in Eq. (3) is replaced with the discrete CE Eq. (14). Concurrently, Guo et al. (2026) considered the same formulation and provided a similar Schrödinger Bridge analysis for CTMCs.

### 3.2. Analysis of GSB on Graphs

We characterize the optimal controlled dynamics associated with the continuity equation and show that they admit a representation in terms of a time-dependent potential function $V_t$. This characterization naturally leads to a discrete analogue of the Hopf-Cole transform, which forms the basis of our subsequent developments.

**Theorem 3.1** (Dual representation). *Under mild regularity assumptions, there exists a time-dependent function $V : [0, 1] \times \mathcal{X} \to \mathbb{R}$ such that the optimal probability path $p_t^\star$ satisfies*

$$\partial_t p_t^\star(x) = \sum_{y \in \mathcal{X}} r_t(x, y) \exp(-V_t(x) + V_t(y)) p_t^\star(y),$$

$$\partial_t V_t(x) = \sum_{y \in \mathcal{X}} r_t(y, x) \exp(-V_t(y) + V_t(x)) - f_t(x, p_t^\star),$$

$$(15)$$

*with boundary conditions $p_0^\star = \mu$ and $p_1^\star = \nu$. In particular, the optimal controlled transition rate $u_t^\star$ is given by*

$$u_t^\star(y, x) = r_t(y, x) \exp(-V_t(y) + V_t(x)). \quad (16)$$

**Remark 3.2.** *The result follows by considering the Lagrangian dual of the GSB objective Eq. (13), where the continuity equation Eq. (14) is enforced via a state-time-dependent adjoint variable $V_t$. Optimizing the resulting dual functional yields the coupled system Eq. (15). Alternatively, $V_t$ can be interpreted as the solution to a discrete Hamilton–Jacobi–Bellman (HJB) equation associated with a stochastic optimal control problem on the graph. Further discussion is deferred to Appendix B.*

**Remark 3.3.** *Importantly, it is underlined that Eq. (16) implies for any $(x,y) \in \mathcal{X} \times \mathcal{X}$, if $r_t(y,x) = 0$, then $u_t^\star(y,x) = 0$. Consequently, $u_t^\star$ satisfies the same graph constraints as $r_t$.*

**Proposition 3.4.** *We define the Hopf-Cole transformation on $\mathcal{G}$ analogous to the dynamic counterpart as*

$$\varphi(t,x) := e^{-V(t,x)}, \qquad \hat{\varphi}(t,x) := \frac{p_t^\star(x)}{\varphi(t,x)}. \qquad (17)$$

*then the Schrödinger potentials $(\varphi_t, \hat{\varphi}_t)$ satisfy*

$$
\begin{aligned}
\partial_t \varphi_t(x) &= -\sum_y r_t(y,x)\,\varphi_t(y) + f_t(x, p_t^\star)\,\varphi_t(x), \\
\partial_t \hat{\varphi}_t(x) &= \sum_y r_t(x,y)\,\hat{\varphi}_t(y) - f_t(x, p_t^\star)\,\hat{\varphi}_t(x),
\end{aligned} \qquad (18)
$$

*with $p_t^\star = \varphi_t \hat{\varphi}_t$, $\forall t \in [0,1]$, and in particular $\nu = \varphi_1 \hat{\varphi}_1$.*

Moreover, the optimal controlled transition rate can be expressed as

$$u_t^\star(y,x) = r_t(y,x)\,\frac{\varphi_t(y)}{\varphi_t(x)}. \qquad (19)$$

Thus, recovering the optimal dynamics reduces to estimating the local ratios $\varphi_t(y)/\varphi_t(x)$ along edges $(x,y) \in \mathcal{E}$.

### 3.3. Generalized IPF for Graph

**Generator and Dynkin's formula** A direct approach to solve the high-dimensional Hopf-Cole equations in Eq. (18) is computationally prohibitive on large graphs. Instead, we adopt a particle-based perspective, discrete analogues of the gIPF and temporal-difference (TD) objectives. Unlike the continuous setting, where these objectives via representations built on Itô calculus, discrete dynamics require different dynamical expressions. For this reason, we consider the Dynkin's formula (Dynkin, 1965; Norris, 1997)

$$\mathbb{E}_{p_{t|s}^u}[h_t(X_t)] = h_s(X_s) + \mathbb{E}_{p_{\cdot|s}^u}\left[\int_s^t \mathcal{A}_\tau^u h(X_\tau)\,d\tau\right], \quad (20)$$

for any test function $h : [0,1] \times \mathcal{X} \to \mathbb{R}$, where $\mathcal{A}_t^u$ denotes the infinitesimal generator associated with the controlled transition rate $u_t$, defined by

$$\mathcal{A}_t^u h(x) := \lim_{\Delta t \to 0} \frac{\mathbb{E}[h_{t+\Delta t}(X_{t+\Delta t}) \mid X_t = x] - h_t(x)}{\Delta t}.$$

The definition of the generator above enables us to introduce, in the next Theorem below, a pair of controlled CTMCs on $\mathcal{G}$ whose generators are supported on on edges $(x,y) \in \mathcal{E}$.

**Proposition 3.5.** *Let $(X_t)_{t \in [0,1]}$ be a CTMC process on $\mathcal{X}$, which evolves under the controlled transition rate $u_t(y,x) = e^{Z_t(y,x)} r_t(y,x)$. For convenience, we define*

$$
\begin{aligned}
Y_t(X_t) &:= \log \varphi(t, X_t), & Z_t(y, X_t) &= Y_t(y) - Y_t(X_t), \\
\widehat{Y}_t(X_t) &:= \log \widehat{\varphi}(t, X_t), & \widehat{Z}_t(y, X_t) &= \widehat{Y}_t(y) - \widehat{Y}_t(X_t).
\end{aligned}
$$

*Then, the pair $(Y_t, \widehat{Y}_t)$ is expressed through the generator $\mathcal{A}_t^u$ as follows*

$$\mathcal{A}_t^u Y(x) = \sum_y r_t e^{Z_t}(Z_t - 1) + f_t(x, p_t^u), \qquad (21)$$

$$\mathcal{A}_t^u \widehat{Y}(x) = \sum_y r_t(x,y) e^{\widehat{Z}_t} + \widehat{Z}_t r_t e^{Z_t} - f_t(x, p_t^u), \quad (22)$$

*where we omit the function arguments when they are $(y,x)$. Moreover, given the backward transition rate $\hat{u}_s(x,y) = e^{\widehat{Z}_s(y,x)} r_s(x,y)$, where $s := 1 - t$*

$$\mathcal{A}_s^{\hat{u}} \widehat{Y}(x) = \sum_y r_s(x,y) e^{\widehat{Z}_s}\left(\widehat{Z}_s - 1\right) + f_s(x, p_s^u), \quad (23)$$

$$\mathcal{A}_s^{\hat{u}} Y(x) = \sum_y r_s e^{Z_s} + Z_s r_s(x,y) e^{\widehat{Z}_s} - f_s(x, p_s^u). \quad (24)$$

**gIPF Objective** It is underlined that Eqs. (21) and (22) are the discrete analogue of the diffusion SB-FBSDE in Eqs. (7) and (8). They characterize the GSB solution by describing the evolution of the $(\log \varphi_t(X_t), \log \widehat{\varphi}_t(X_t))$ along the trajectories, thus avoiding time-expanded global solves over all nodes and time indices. We leverage these equations to derive a *data-driven log-likelihood maximization objective*, by expanding the SB potentials along CTMC trajectories:

$$
\begin{aligned}
-\log \mu(X_0) &\underset{(20)}{=} \mathbb{E}_t \mathbb{E}_{p_{\cdot|0}^Z}\big[\mathcal{A}_t^u \log p_t^Z(X_t)\big] + C \\
&= \mathbb{E}_t \mathbb{E}_{p_{\cdot|0}^Z}\big[\mathcal{A}_t^u Y_t(X_t) + \mathcal{A}_t^u \widehat{Y}_t(X_t)\big] + C, \quad (25)
\end{aligned}
$$

where $C = -\mathbb{E}_{p_{1|0}^Z}[\log \nu(X_1)]$, and $p^Z := p^u$ (resp. $p^{\widehat{Z}} := p^{\hat{u}}$). This yields the discrete gIPF loss.

**Proposition 3.6.** *Assume the controlled process $(X_t)_{t \in [0,1]}$ on $\mathcal{X}$, sampled with $u_t$ in Eq. 19. Expansion of the generator terms in Eq. (25) gives the forward IPF objective.*

$$
\begin{aligned}
\mathcal{L}_{\mathrm{IPF}}^Z(\widehat{Z}) := \mathbb{E}_{p^Z}\Bigg[ \int_0^1 \sum_{y \in \mathcal{X}} \Big( & r_t(X_t, y) e^{\widehat{Z}_t(y, X_t)} + \\
& r_t(y, X_t) e^{Z_t(y, X_t)}(-1 + Z_t(y, X_t) + \widehat{Z}_t(y, X_t)) \Big) dt \Bigg]. \quad (26)
\end{aligned}
$$

*Similarly for the time reversed dynamics, the corresponding backward IPF objective admits a similar expansion*

$$\mathcal{L}_{\text{IPF}}^{\widehat{Z}}(Z) = \mathbb{E}_{p^{\hat{Z}}}\bigg[ \int_0^1 \sum_{y \in \mathcal{X}} \Big( r_s(y, \tilde{X}_s) e^{Z_s(y, \tilde{X}_s)} +$$
$$r_s(\tilde{X}_t, y) e^{\widehat{Z}_s(y, \tilde{X}_s)} \big( -1 + Z_s(y, \tilde{X}_t) + \widehat{Z}_s(y, \tilde{X}_s) \big) \Big) \, \mathrm{d}s \bigg].$$
(27)

**Temporal-difference (TD) objective** A key observation is in the IPF objective that the state cost term appears with opposite signs: in $\mathcal{A}^u Y(t, x)$ its contribution is $+f_t$, whereas in $\mathcal{A}^u \widehat{Y}(t, x)$ it enters as $-f_t$, thus, it cancels in the sum $\mathcal{A}^u_t Y(x) + \mathcal{A}^u \widehat{Y}_t(x)$. Consequently, solely minimizing the IPF, in the case of nontrivial running costs, is insufficient, as it can enforce the endpoint marginals, but does not explicitly enforce consistency with the state-/distribution-dependent running cost.

To recover the correct cost-aware dynamics, we additionally enforce *local generator consistency*, through temporal-difference (TD) loss, penalizing violations of Eqs. (21) and (22) along the trajectories. The increments of $(Y_t, \widehat{Y}_t)$ over a short interval must agree–in expectation– with the action of the controlled generator, which is precisely where the running cost appears. More specifically, for the forward CTMC $(X_t)_{t \in [0,1]}$ sampled via $u_t(y, x) = e^{Z_t(y,x)} r_t(y, x)$, application of Dynkin's formula Eq. (20), on $[t, t + \Delta t]$ for $\widehat{Y}_t$ enforces local generator consistency, as the one-step increment $\delta \widehat{Y}_t := \widehat{Y}_{t+\Delta t}(X_{t+\Delta t}) - \widehat{Y}_t(X_t)$ should match the generator-predicted increment $\mathcal{A}_t^{u^\theta} \widehat{Y}(X_t) \Delta t$. Similarly, for the time-reversed process $\tilde{X}_s$, we apply Eq. (20) on $[s - \Delta s, s]$ for $Y_t$ and enforce the same consistency condition. Consequently, we obtain the following forward-backward pair of TD losses as the squared residual of this condition:

$$\mathcal{L}_{\text{TD}}^Z(\widehat{Y}) = \mathbb{E}_{y \sim p_{t+\Delta t|t}^Z(\cdot|x)} \Big[ \| \delta \widehat{Y}_t - \mathcal{A}_t^u \widehat{Y} \Delta t \|^2 \Big] \quad (28)$$

$$\mathcal{L}_{\text{TD}}^{\widehat{Z}}(Y) = \mathbb{E}_{y \sim p_{s-\Delta s|s}^{\hat{Z}}(\cdot|x)} \Big[ \| \delta Y_s - \mathcal{A}_s^u Y \Delta s \|^2 \Big], \quad (29)$$

with $\widehat{Y}_0 = \log p_0 - Y_0$ for the forward process, and $Y_T = \log p_1 - \widehat{Y}_1$ for the backward. In practice, our training alternates between minimizing the IPF objectives – to match endpoint marginals – and minimizing the TD objectives – to enforce local generator consistency with the running cost– producing a scheme that simultaneously satisfies boundary constraints and cost-consistent path-space optimality.

**Training Objective and Parameterization** Algorithm 1 summarizes our training scheme. In practice, we parameterize the log-potentials $(\theta, \phi)$: $Y_t^\theta(X_t) \approx$

---

**Algorithm 1** Generalized Schrödinger Bridge on Graphs

1: **Input:** graph $\mathcal{G}$, reference rates $r_t$, endpoint marginals $p_0, p_1$, cost $f_t(x, p_t)$
2: Initialize parameters $\theta, \phi$; choose time grid $\{t_k\}_{k=0}^K$, $\Delta t = 1/K$
3: **for** iterations $m = 1, 2, \dots$ **do**
4:     **Forward rollouts:** sample $X_0^{(i)} \sim p_0$; simulate CTMC under $u_t^\theta$ in Eq. (30)
5:     **Update** $\phi$: minimize $\mathcal{L}_{\text{IPF}}^Z(\widehat{Z}^\phi) + \lambda_{\text{TD}} \mathcal{L}_{\text{TD}}^Z(\widehat{Y}^\phi)$ from Eqs. (26) and (28) using forward-sampled trajectories
6:     **Backward rollouts:** sample $\tilde{X}_0^{(i)} \sim p_1$; simulate time-reversed dynamics under $\hat{u}^\phi$ in Eq. (31)
7:     **Update** $\theta$: minimize $\mathcal{L}_{\text{IPF}}^{\widehat{Z}}(Z^\theta) + \lambda_{\text{TD}} \mathcal{L}_{\text{TD}}^{\widehat{Z}}(Y^\theta)$ from Eqs. (27) and (29) using backward trajectories
8: **end for**

---

$\log \varphi_t(X_t)$, $\widehat{Y}_t^\phi(X_t) \approx \log \widehat{\varphi}_t(X_t)$, from which the controlled transition rates are given by

$$u_t^\theta(y, x) = r_t(y, x) \exp\big( Z_t^\theta(y, x) \big), \quad (30)$$
$$\hat{u}_t^\phi(y, x) = r_t(y, x) \exp\big( \widehat{Z}_t^\phi(y, x) \big) \quad (31)$$

. Finally, the loss that we use in practice is as follows

$$\mathcal{L} = \mathcal{L}_{\text{IPF}}^Z(\widehat{Z}^\phi) + \mathcal{L}_{\text{IPF}}^{\widehat{Z}}(Z^\theta) + \lambda_{\text{TD}} \Big( \mathcal{L}_{\text{TD}}^Z(\widehat{Y}^\phi) + \mathcal{L}_{\text{TD}}^{\widehat{Z}}(Y^\theta) \Big),$$

where $\lambda_{\text{TD}} \geq 0$ weighting the TD regularization relative to the IPF terms.

## 4. Experiments

### 4.1. Supply Chain

We study the capacity of our model in navigating complex and sparse dynamics on a graph. We model a real-world supply-demand network (Kovács, 2015), as a directed weighted graph $\mathcal{G}$ with $N = 9559$ nodes, where each node represents a facility/location

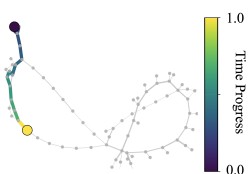

*Figure 2.* Visualization of a 65-node subgraph of the supply chain setup.

(e.g., supplier, depot, port, retailer), and each directed edge represents an admissible shipment arc, as illustrated in Figure 2. The sparse CTMC reference generator encodes the network topology and a nominal routing bias via route costs and capacities. Given boundary marginals $(p_0, p_1)$ (e.g., representing supply and demand), we solve a finite-horizon transport problem, augmenting the objective with a running cost $f_t(x, p_t)$ that penalizes congestion, thus discouraging high-occupancy nodes while still matching $(p_0, p_1)$. Additional details are left for Appendix C.2.

*Table 2.* Supply-chain comparison between our GSBoG and $\mathcal{W}1_{\text{flow}}$, GrSB and Attr. Flow in Total Variation (TV), mean congestion of the 100 most occupied nodes, peak node occupancy, and maximum edge flow over capacity. (Lower is better)

| Method | TV ↓ | Mean congestion ↓ | Peak occupancy ↓ | Max flow/cap ↓ |
|---|---|---|---|---|
| GrSB | — | — | — | — |
| Attr. Flow | 0.27 | 84.09 | 1033.72 | 1.59 |
| $\mathcal{W}1_{\text{flow}}$ | **0.03** | 178.19 | 1807.66 | 1.41 |
| GSBoG | **0.03** | **21.13** | **271.49** | **0.64** |

We compare GSBoG against the dynamic graph Schrödinger Bridge (GrSB; (Chow et al., 2022)) and two graph-native dynamic flow baselines: (i) an Attraction-Flow that biases transitions toward the target, and (ii) a dynamic $\mathcal{W}_1$ minimum-cost flow baseline on the graph ($\mathcal{W}1_{\text{flow}}$). Since flow methods yield fluxes rather than a stepwise routing policy, we employ a Markovian embedding to convert them into a stochastic transition kernel and simulate trajectories. All metrics reported below are therefore *rollout-based*, using the same horizon and rollout budget for all methods. We report: (i) terminal Total Variation, (ii) mean congestion of the 100 most occupied nodes, (iii) peak node occupancy, and (iv) the maximum ratio of induced edge-flow to capacity.

Table 2 shows that terminal matching is not the main difficulty; the challenge is achieving it without inducing infeasible, highly crowded intermediate behavior. GrSB did not complete on this graph due to memory exhaustion issues, even after reducing the horizon and rollout count, highlighting scalability limitations at this scale. Among the flow-based baselines, the induced rollouts exhibit substantially higher congestion and large realized capacity overload. In particular, $\mathcal{W}1_{\text{flow}}$ attains near-perfect terminal matching but routes most mass through a small set of bottleneck edges, resulting in severe intermediate crowding. In contrast, GSBoG maintains near-optimal terminal matching while dramatically suppressing intermediate congestion. Importantly, GSBoG is the only method in our study whose induced trajectories remain within edge capacities at this scale. Although capacity satisfaction is not enforced as a hard constraint, it is achieved implicitly, through the congestion penalty which incentivizes policies that spread mass through less crowded routes, avoiding congestion.

Importantly, Figure 3b highlights the strong impact of incorporating a non-trivial application-specific state cost. In particular, including a congestion cost ($f \neq 0$) further suppresses crowding, as it leads to policies which spread mass more evenly, rather than collapsing onto a few bottleneck nodes, while preserving the endpoint matching. Notably, the congestion-aware policy nearly halves the peak occupancy relative to the $f = 0$ variant. This mitigates congestion, rendering the congestion-aware solutions more robust to

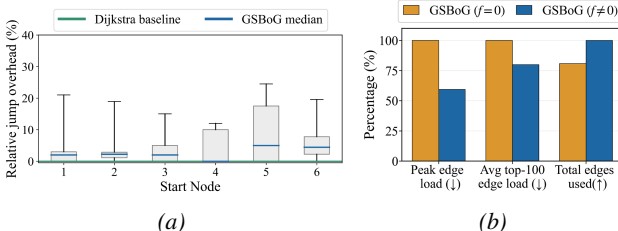

*(a)*        *(b)*

*Figure 3.* **Left:** Distribution of relative path length overhead of GSBoG policy compared to Dijkstra. **Right:** Edge-utilization summary between GSBoG without ($f = 0$) and with ($f \neq 0$) state-cost: Peak edge load, average load of the top 100 most used edges by total number of particle traversals and total number of edges used.

*Table 3.* Assignment task metrics

| $n$ | Optimal Cost | Assigned Cost | Mass on opt. selection | Row Entropy | Accuracy |
|---|---|---|---|---|---|
| 6 | 160 | 160 | 97.3% | 0.07 | 100% |
| 8 | 205 | 205 | 94.2% | 0.13 | 100% |
| 10 | 239 | 239 | 87.6% | 0.16 | 100% |
| 20 | 538 | 546 | 85.0% | 0.17 | 90% |

disruptions. Lastly, Figure 3a shows that GSBoG remains near shortest-path behavior, incurring only a modest jump overhead on average across all sources.

### 4.2. Assignment Task

We test GSBoG on a balanced assignment task with $n$ supply nodes $A_1, \ldots, A_n$ and $n$ demand nodes $B_1, \ldots, B_n$, with $n \in \{6, 8, 10, 20\}$. The ground-truth plan $\pi^\star$ is the minimum-cost assignment under $C \in \mathbb{R}^{n \times n}$ that matches marginals $p_0, p_1$. To encode pairwise costs using state penalties, we build a directed graph with intermediate nodes $E_{xy}$ for each $(x, y)$ and edges $A_x \to E_{xy} \to B_y$, assigning $f(E_{xy}) = C_{xy}$ and $f(A_x) = f(B_y) = 0$. Running GSBoG on this graph yields controlled dynamics whose trajectories define the learned assignment $\hat{\pi}$ from the expected flux on edges $A_x \to E_{xy}$ and compare it with $\pi^\star$. More details are left for Appendix C.1.

Figure 4 shows the bipartite assignment graph, along with the heatmap of the soft plan $\hat{\pi}$ together with the corresponding hard assignment (red boxes), overlaid with the oracle solution $\pi^\star$ obtained by solving the static OT (orange markers). Quantitatively, we see that for $n \leq 10$ the hard assignment matches the OT solution exactly (100% overlap) while remaining fairly confident with high percentage of mass on the optimal edges. For $n = 20$, the learned policy still places 85% mass on the correct edges, and the hard overlap drops to 90%, indicating that only one assignment pair is misplaced. Importantly, notice that Table 3 suggests that this misplacement increased the cost only marginally, implying that the selected alternative is near-optimal and that GSBoG

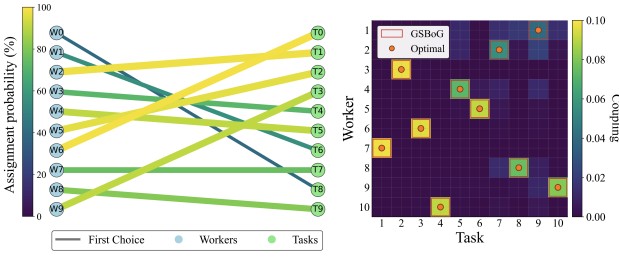

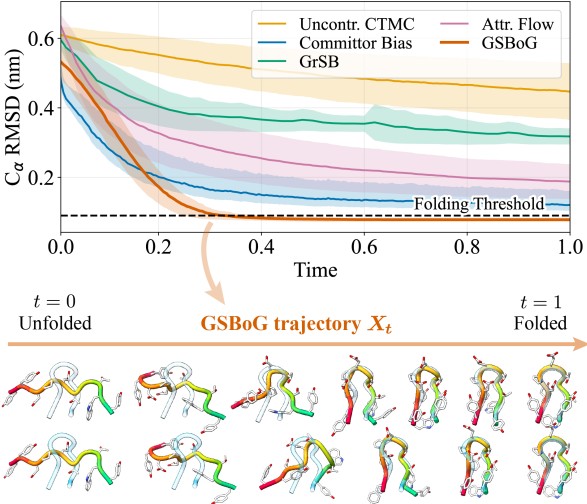

*Figure 4.* **Left:** Assignment probabilities on the worker–task bipartite graph. Edge thickness and color encode the probability of assigning each worker $W_i$ to task $T_j$ under the learned policy. **Right:** Heatmap shows the learned coupling (GSBoG) between workers (rows) and tasks (columns). Orange markers denote the optimal assignment.

*Table 4.* Chignolin folding comparison between our GSBoG and Uncontrolled CTMC, Committor-Guided bias, GrSB and Attr. Flow in folding rate and energy barrier

| Method | Energy barrier $(k_B T_{sim})$ | Fold rate (%) |
|---|---|---|
| Uncontr. CTMC | 6.94 | 0.55 |
| Committor-guided bias | 6.59 | 71.69 |
| GrSB | 5.43 | 25.11 |
| Attr. Flow | 7.32 | 62.08 |
| GSBoG ($f = 0$) | 1.80 | **99.70** |
| GSBoG ($f \neq 0$) | **1.39** | 99.36 |

remains close to the ground truth optimal solution.

### 4.3. Discretized Molecular Dynamics

Finally, we evaluate GSBoG for steering rare-event molecular kinetics in a discretized configurational space represented by a Markov state model (MSM; Husic & Pande (2018)). In this task, the goal is twofold: increase the probability of observing a kinetically suppressed transition within a fixed time horizon, while avoiding thermodynamically implausible regions of the state space. As a test system, we consider the folding of chignolin in water, a 10-residue miniprotein that undergoes rare but reversible transitions between extended (unfolded) and $\beta$-hairpin (folded) structures, which is a standard model system for protein folding dynamics (Lindorff-Larsen et al., 2011).

An MSM here is a Markov chain on a finite set of $|\mathcal{X}| = 500$ configurational microstates obtained by clustering equilibrated molecular dynamics trajectories. From these trajectories, the microstate populations $\pi_i$, the corresponding free energies of each microstate $F_i/k_B T_{sim} = -\log \pi_i$ at simulation temperature $T_{sim}$ and a reference rate generator are estimated. Although MSMs are a standard tool for analyzing molecular dynamics, identifying dominant transition pathways between metastable basins remains difficult due

*Figure 5.* **Top:** Mean chignolin $C_\alpha$ RMSD across trajectories from different methods, with shaded areas indicating standard deviation. **Bottom:** Trajectory samples generated by GSBoG (colored) overlaid on the native chignolin structure (transparent). GSBoG correctly *folds* the structure over time.

to the extreme rarity of transitions: an uncontrolled CTMC initialized in the unfolded basin distribution $\mu$ reaches the folded basin distribution $\nu$ within horizon $T = 200$ with probability below 1% (Table 4). We formulate this as a GSB problem on the MSM graph, with controlled transport from $\mu$ to $\nu$ under reference dynamics. Furthermore, we introduce a running cost $f$ using the free energy of each microstate, which biases the controlled dynamics toward thermodynamically stable regions of the MSM and thus encourages physically plausible folding pathways. Additional experiment details are provided in Appendix C.3.

In Table 4, all guided baselines increase the folding probability, but there is a gap between simply reaching the endpoint and doing so via low-barrier intermediates. Committor-guided bias and Attraction Flow raise the fold rate to 60–70%, respectively, yet still traverse high barriers, indicating that successful trajectories often pass through high energy regions. GrSB improves over the uncontrolled prior (folding 25% of the time and reducing the barrier to $5.43\,k_B T_{sim}$) but still falls short of reliably inducing folding at this horizon. In contrast, GSBoG attains both high folding probability and substantially reduced barriers: the variant with $f = 0$ reaches a 99.7% fold rate with a $1.80\,k_B T_{sim}$ barrier, and incorporating the free energy running cost further lowers the barrier to $1.39\,k_B T_{sim}$ while maintaining a high fold rate. In addition, by measuring the geometric deviation (RMSD, Eq. (97)) from the native folded structure along trajectories using cluster representative structures (Fig. 5), we confirm that GSBoG trajectories follow smooth folding pathways and terminate within the folded state RMSD

threshold. Overall, this shows that GSBoG is effective for rare-event steering: endpoint constraints ensure folding, and the free energy cost $f$ further biases trajectories toward thermodynamically plausible pathways.

### 4.4. Scalability of GSBoG

A central advantage of GSBoG is that it avoids global time-expanded graph solvers. Rather than optimizing over all node marginals or time-indexed flux variables, GSBoG learns trajectory-level CTMC policies whose controlled rates are evaluated only on local graph neighborhoods. Since $u_t^\star(y, x) = r_t(y, x)\phi_t(y)/\phi_t(x)$ for $(x, y) \in E$, and $u_t^\star(y, x) = 0$ otherwise, each particle at node $x$ only needs to consider $y \in \mathcal{N}(x)$. Thus, the per-step computation is governed by the local branching factor $\Delta$, rather than the total number of nodes $n$.

This locality is particularly beneficial on sparse graphs. GSBoG stores and updates quantities on the sparse edge set and sampled trajectories, whereas global dynamic OT and graph SB solvers operate over all nodes, edges, and time layers. In the supply-chain graph, for example, $n = 9559$ while the maximum degree is only 6. Hence each GSBoG particle interacts with at most six neighbors per step, explaining its empirical scalability and its ability to remain feasible when global solvers become memory-limited or fail to complete.

**Runtime and memory.** Figure 6 reports memory usage (top) and wall-clock time (bottom) for Dynamic OT and GSBoG across four scaling axes: number of nodes, edge density, time steps, and rollout budget. Entries marked "N/A" denote settings where Dynamic OT did not complete. Across all regimes where both methods run, GSBoG is substantially more resource efficient. At $5 \times 10^4$ nodes, for instance, GSBoG uses only 1.4 GB and finishes in roughly 1 hour, compared with 22.0 GB and about 4.6 hours for Dynamic OT. Similar reductions appear when varying edge density, horizon length, and rollout budget: GSBoG consistently uses less memory and less time, while Dynamic OT either incurs much larger costs or becomes infeasible.

**Scaling behavior.** The four panels highlight the difference between local trajectory-based learning and global time-expanded optimization. As the graph grows from $5 \times 10^4$ to $10^6$ nodes, Dynamic OT becomes infeasible after the smallest instance, whereas GSBoG remains executable up to $10^6$ nodes, with memory increasing gradually. Increasing the number of edges shows the same trend: Dynamic OT fails at the densest $5 \times 10^6$-edge setting, while GSBoG still completes. Along the time dimension, Dynamic OT scales poorly with the number of steps and fails at 400 steps, whereas GSBoG maintains low memory and moderate runtime. Finally, the rollout-budget ablation shows that GSBoG can trade computation for a more accurate particle

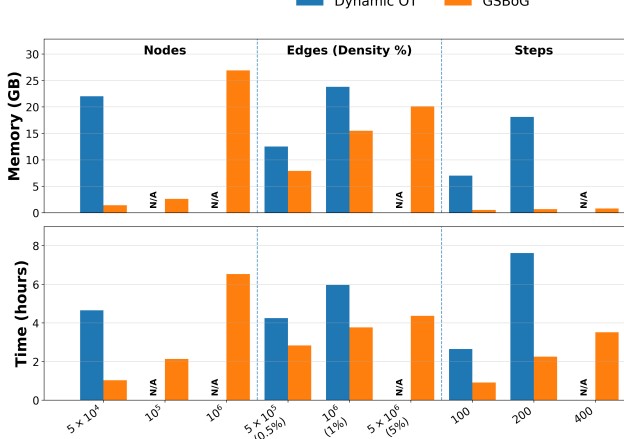

*Figure 6.* Scalability of Dynamic OT and GSBoG across varying numbers of nodes, edge densities, time steps, and rollout budgets. GSBoG consistently uses less memory and runtime than Dynamic OT, and remains feasible in settings where Dynamic OT becomes infeasible (marked N/A).

approximation. Increasing the budget from $10^3$ to $4 \times 10^3$ increases runtime, but memory remains below 1 GB. Overall, these results support the main computational picture: global dynamic OT and graph SB methods operate over the full time-expanded graph, whereas GSBoG operates on sampled local trajectories. This makes GSBoG particularly suitable for large sparse graphs, where each node has only a small number of feasible neighbors.

### 4.5. Limitations

GSBoG currently assumes that the underlying graph topology is fixed and fully known, since both the reference CTMC generator and the learned controlled rates are defined on the admissible edge set. While extensions to time-varying graphs may be possible through time-dependent generators and supports, handling partially observed, uncertain, or dynamically evolving connectivity remains a nontrivial direction, as rollout feasibility and loss evaluation require access to the valid transitions. We therefore leave the extension of GSBoG to uncertain, partially observed, and time-evolving graph structures as important future work.

## 5. Discussion

In this work, we present a first graph-native Schrödinger-bridge framework for optimal stochastic policies that transport mass between prescribed marginals while accounting for network structure and operational constraints such as congestion and capacity. This work takes a step towards data-driven, principled approaches for transport on large networks, paving new avenues for integrating modern learning-based dynamics with optimal-transport-inspired control in complex networked systems.

## Acknowledgments

We thank Guan-Horng Liu for his assistance with the theoretical analysis, as well as for his insightful discussions and comments on the manuscript. This research is partially supported by the DARPA AIQ program through the DARPA CMO contract number HR00112520010. We would like to thank Dr. Pat Shafto, AIQ Program Manager, for useful technical discussions.

## Impact Statement

This paper presents work whose goal is to advance the field of machine learning. There are many potential societal consequences of our work, none of which we feel must be specifically highlighted here.

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

# A. Proofs

## A.1. Proof of Theorem 3.1

**Theorem A.1.** *Under mild regularity assumptions, there exists a time-dependent function $V : [0, 1] \times \mathcal{X} \to \mathbb{R}$ such that the optimal probability path $p_t^\star$ satisfies*

$$\partial_t p_t^\star(x) = \sum_{y \in \mathcal{X}} r_t(x, y) \exp(-V_t(x) + V_t(y)) p_t^\star(y),$$

$$\partial_t V_t(x) = \sum_{y \in \mathcal{X}} r_t(y, x) \exp(-V_t(y) + V_t(x)) - f_t(x, p_t^\star), \tag{32}$$

*with boundary conditions $p_0^\star = \mu$ and $p_1^\star = \nu$. In particular, the optimal controlled transition rate $u_t^\star$ is given by*

$$u_t^\star(y, x) = r_t(y, x) \exp\big(-V_t(y) + V_t(x)\big). \tag{33}$$

*Proof.* Let $\mathcal{X}$ be a finite (or countable) state space. We use the convention that $u_t(y, x)$ is the jump rate *from $x$ to $y$* (so $x$ is the column index). Thus the controlled generator satisfies

$$u_t(x, x) = -\sum_{y \in \mathcal{X},\ y \neq x} u_t(y, x), \qquad r_t(x, x) = -\sum_{y \in \mathcal{X},\ y \neq x} r_t(y, x),$$

where $r_t$ is the reference generator. We consider the Generalized Schrödinger Bridge problem on $\mathcal{X}$, where we have substituted for the $KL$

$$\inf_u \int_0^1 \sum_{x \in \mathcal{X}} \sum_{\substack{y \in \mathcal{X} \\ y \neq x}} \left[ -u_t(y, x) + r_t(y, x) + u_t(y, x) \log\left(\frac{u_t(y, x)}{r_t(y, x)}\right) + f_t(x, p_t) \right] p_t(x)\, dt \tag{34}$$

$$\text{s.t. } \partial_t p_t(x) = \sum_{y \in \mathcal{X}} u_t(x, y)\, p_t(y) \tag{35}$$

$$p_0 = \mu, \qquad p_1 = \nu. \tag{36}$$

We introduce a Lagrangian multiplier $V : [0, 1] \times \mathcal{X} \to \mathbb{R}$, which recasts the constrained optimization problem in Eqs. (34) and (35) as follows:

$$\mathcal{L}(u, p, V) = \int_0^1 \sum_{x \in \mathcal{X}} \sum_{\substack{y \in \mathcal{X} \\ y \neq x}} \left[ -u_t(y, x) + r_t(y, x) + u_t(y, x) \log\left(\frac{u_t(y, x)}{r_t(y, x)}\right) + f_t(x, p_t) \right] p_t(x)\, dt$$

$$- \int_0^1 \sum_{x \in \mathcal{X}} V_t(x) \left[ \partial_t p_t(x) - \sum_{y \in \mathcal{X}} u_t(x, y) p_t(y) \right] dt. \tag{37}$$

where $L$ is the Lagrangian of our problem in Eqs. (34) and (35).

Subsequently, we perform integration by parts to $\sum_{x \in \mathcal{X}} V_t(x)\, \partial_t p_t(x)$, and rewrite $\sum_{x \in \mathcal{X}} \sum_{y \in \mathcal{X}} V_t(x)\, u_t(x, y)\, p_t(y)$, as follows:

- $\sum_{x \in \mathcal{X}} V_t(x)\, \partial_t p_t(x)$

$$\int_0^1 \sum_{x \in \mathcal{X}} V_t(x)\, \partial_t p_t(x)\, dt = -\int_0^1 \sum_{x \in \mathcal{X}} (\partial_t V_t(x))\, p_t(x)\, dt + \sum_{x \in \mathcal{X}} \left[ V_1(x)\, p_1(x) - V_0(x)\, p_0(x) \right]$$

$$= -\int_0^1 \sum_{x \in \mathcal{X}} (\partial_t V_t(x))\, p_t(x)\, dt + \sum_{x \in \mathcal{X}} \left[ V_1(x)\, \nu(x) - V_0(x)\, \mu(x) \right]. \tag{38}$$

- $\sum_{x \in \mathcal{X}} \sum_{y \in \mathcal{X}} V_t(x) \, u_t(x, y) \, p_t(y)$

$$\sum_{x \in \mathcal{X}} \sum_{y \in \mathcal{X}} V_t(x) \, u_t(x, y) \, p_t(y) = \sum_{y \in \mathcal{X}} \sum_{x \in \mathcal{X}} V_t(x) \, u_t(x, y) \, p_t(y)$$

$$= \sum_{x \in \mathcal{X}} \sum_{\substack{y \in \mathcal{X} \\ y \neq x}} V_t(y) \, u_t(y, x) \, p_t(x) \; + \; \sum_{x \in \mathcal{X}} V_t(x) \, u_t(x, x) \, p_t(x)$$

$$= \sum_{x \in \mathcal{X}} \sum_{\substack{y \in \mathcal{X} \\ y \neq x}} \Big( V_t(y) - V_t(x) \Big) \, u_t(y, x) \, p_t(x) \tag{39}$$

where we used $u_t(x, x) = - \sum_{y \neq x} u_t(y, x)$.

Plugging Eqs. (38) and (39) into Eq. (37) yields

$$\mathcal{L}(u, p, V) = \int_0^1 \sum_{x \in \mathcal{X}} \sum_{\substack{y \in \mathcal{X} \\ y \neq x}} \Big[ -u_t(y, x) + r_t(y, x) + u_t(y, x) \log\Big( \frac{u_t(y, x)}{r_t(y, x)} \Big) + f_t(x, p_t) \Big] p_t(x) \, dt$$

$$+ \int_0^1 \sum_{x \in \mathcal{X}} (\partial_t V_t(x)) \, p_t(x) \, dt - \sum_{x \in \mathcal{X}} \Big[ V_1(x) \nu(x) - V_0(x) \mu(x) \Big]$$

$$+ \int_0^1 \sum_{x \in \mathcal{X}} \sum_{\substack{y \in \mathcal{X} \\ y \neq x}} \Big( V_t(y) - V_t(x) \Big) \, u_t(y, x) \, p_t(x) \, dt. \tag{40}$$

We invoke strong duality

$$\sup_V \inf_{(u, p)} \mathcal{L}(u, p, V)$$

and focus on the terms depending on $u_t(y, x)$ inside Eq. (40):

$$-u + u \log\Big( \frac{u}{r} \Big) + (V_t(y) - V_t(x)) u.$$

Taking the derivative with respect to $u_t$ yields

$$0 = \frac{\partial}{\partial u} \Big[ -u + u \log\Big( \frac{u}{r} \Big) + (V_t(y) - V_t(x)) u \Big] = \log\Big( \frac{u}{r} \Big) + V_t(y) - V_t(x),$$

$$\implies \quad u_t^\star(y, x) = r_t(y, x) \, \exp\big( -V_t(y) + V_t(x) \big). \tag{41}$$

Therefore, the optimal control of the GSB problem on $\mathcal{X}$ is given by

$$u_t^\star(y, x) = r_t(y, x) \, \exp\big( -V_t(y) + V_t(x) \big). \tag{42}$$

Finally, plugging $u^\star$ back into the Lagrangian and substituting Eq. (41) into Eq. (40) yields

$$\mathcal{L}(u^\star, p, V) = \int_0^1 \sum_{x \in \mathcal{X}} \sum_{\substack{y \in \mathcal{X} \\ y \neq x}} \Big[ \Big( 1 - e^{-V_t(y) + V_t(x)} \Big) r_t(y, x) + f_t(x, p_t) \Big] p_t(x) \, dt$$

$$+ \int_0^1 \sum_{x \in \mathcal{X}} (\partial_t V_t(x)) \, p_t(x) \, dt - \sum_{x \in \mathcal{X}} \Big[ V_1(x) \nu(x) - V_0(x) \mu(x) \Big]$$

$$= \int_0^1 \sum_{x \in \mathcal{X}} \Big[ \partial_t V_t(x) + \sum_{\substack{y \in \mathcal{X} \\ y \neq x}} \Big( 1 - e^{-V_t(y) + V_t(x)} \Big) r_t(y, x) + f_t(x, p_t) \Big] p_t(x) \, dt$$

$$+ \sum_{x \in \mathcal{X}} \Big[ V_0(x) \mu(x) - V_1(x) \nu(x) \Big]. \tag{43}$$

For $\inf_p \mathcal{L}(u^\star, p, V)$ to be nontrivial (finite), the bracketed term in Eq. (43) must vanish (pointwise in $(t, x)$), i.e.

$$0 = \partial_t V_t(x) + \sum_{\substack{y \in \mathcal{X} \\ y \neq x}} r_t(y, x) - \sum_{\substack{y \in \mathcal{X} \\ y \neq x}} e^{V_t(x) - V_t(y)} r_t(y, x) + f_t(x, p_t). \tag{44}$$

Using $r_t(x, x) = -\sum_{y \neq x} r_t(y, x)$, equivalently

$$\partial_t V_t(x) = \sum_{y \in \mathcal{X}} e^{V_t(x) - V_t(y)} r_t(y, x) - f_t(x, p_t). \tag{45}$$

Meanwhile, the continuity equation under the optimal rates is

$$\partial_t p_t^\star(x) = \sum_{y \in \mathcal{X}} u_t^\star(x, y) p_t(y) = \sum_{y \in \mathcal{X}} e^{-V_t(x) + V_t(y)} r_t(x, y) p_t^\star(y), \qquad p_0 = \mu, \ \ p_1 = \nu. \tag{46}$$

Yielding the coupled "continuity–dual" system is Eqs. (45) and (46). □

## A.2. Proof of Proposition 3.4

**Proposition A.2.** *We define the Hopf-Cole transformation on $\mathcal{G}$ analogous to the dynamic counterpart as*

$$\varphi(t, x) := e^{-V(t, x)}, \qquad \hat{\varphi}(t, x) := \frac{p_t^\star(x)}{\varphi(t, x)}. \tag{47}$$

*then the Schrödinger potentials $(\varphi_t, \hat{\varphi}_t)$ satisfy*

$$\partial_t \varphi_t(x) = -\sum_y r_t(y, x) \varphi_t(y) + f_t(x, p_t^\star) \varphi_t(x),$$
$$\partial_t \widehat{\varphi}_t(x) = \sum_y r_t(x, y) \widehat{\varphi}_t(y) - f_t(x, p_t^\star) \widehat{\varphi}_t(x), \tag{48}$$

*with $p_t^\star = \varphi_t \widehat{\varphi}_t$ for all $t \in [0, 1]$, and in particular $\nu = \varphi_1 \widehat{\varphi}_1$.*

*Proof.* Define the Hopf-Cole transformation on the Schrödinger potentials

$$\varphi_t(x) = e^{-V_t(x)}, \qquad \widehat{\varphi}_t(x) = \frac{p_t(x)}{\varphi_t(x)} \quad \Longleftrightarrow \quad p_t(x) = \varphi_t(x) \widehat{\varphi}_t(x).$$

**Equation for $\varphi$.** By the chain rule, $\partial_t \varphi_t(x) = -\varphi_t(x) \partial_t V_t(x)$, we can obtain the following differential equations pertaining to the Schrödinger potentials $\varphi$.

$$
\begin{aligned}
\partial_t \varphi(t, x) &= -\partial_t V_t(x) e^{-V(t, x)} \\
&= \Big[\sum_{y \neq x} r_t(y, x)(1 - e^{V_t(x) - V_t(y)}) + f_t(x_t, p_t)\Big] \varphi(t, x) \\
&= -\sum_{y \neq x} r_t(y, x)(e^{V_t(x) - V_t(y)} - 1) \varphi(t, x) + f_t(x_t, p_t) \varphi(t, x) \\
&= -\sum_{y \neq x} r_t(y, x)\Big(\frac{\varphi(t, y)}{\varphi(t, x)} - 1\Big) \varphi(t, x) + f_t(x_t, p_t) \varphi(t, x) \\
&= -\Big(\sum_{y \neq x} r_t(y, x) \varphi(t, y) - \sum_{y \neq x} r_t(y, x) \varphi(t, x)\Big) + f_t(x_t, p_t) \varphi(t, x) \\
&= -\sum_y \varphi(t, y) r_t(y, x) + f_t(x_t, p_t) \varphi(t, x)
\end{aligned}
\tag{49}
$$

where we use the standard convention $r_t(x, x) = -\sum_{y \neq x} r_t(y, x)$ to absorb diagonal terms into the full sum over $y \in \mathcal{X}$.

**Equation for $\widehat{\varphi}$.** Similarly, for the $\widehat{\varphi}(t,x)$, after differentiation $\widehat{\varphi}_t(x) = p_t(x)/\varphi_t(x)$:

$$\partial_t \widehat{\varphi}_t(x) = \frac{\partial_t p_t(x)}{\varphi_t(x)} - \frac{p_t(x)}{\varphi_t(x)^2}\,\partial_t \varphi_t(x).$$

Using the forward equation Eq. (46) for $p_t$ and Eq. (49) for $\varphi_t$, and recalling $p_t = \varphi_t \widehat{\varphi}_t$, a direct cancellation gives

$$
\begin{aligned}
\partial_t \widehat{\varphi}_t(x) &= \partial_t\left(\frac{p_t^*(x)}{\varphi_t(x)}\right) = \frac{1}{\varphi_t(x)}\sum_y u_t^*(x,y)p_t(y) - \frac{p_t^*(x)}{\varphi_t^2(x)}\partial_t\varphi_t(x)\\
&= \frac{1}{\varphi_t(x)}\left[\sum_{y\neq x} u_t(x,y)p_t(y) + u_t(x,x)p_t(x)\right] - \frac{p_t(x)}{\varphi_t^2(x)}\left[-\sum_y \varphi_t(y)r_t(y,x) + f_t(x_t,p_t)\varphi_t(x)\right]\\
&= \frac{1}{\varphi_t(x)}\left[\sum_{y\neq x} u_t(x,y)p_t(y) - \sum_{y\neq x} u_t(y,x)p_t(x)\right] + \frac{p_t(x)}{\varphi_t^2(x)}\sum_y \varphi_t(y)r_t(y,x) - f_t(x_t,p_t)\varphi_t(x)\frac{p_t(x)}{\varphi_t^2(x)}\\
&= \frac{1}{\varphi_t(x)}\left[\sum_{y\neq x} r_t(x,y)\frac{\varphi_t(x)}{\varphi_t(y)}p_t(y) - \sum_{y\neq x} r_t(y,x)\frac{\varphi_t(y)}{\varphi_t(x)}p_t(x)\right] + \frac{p_t(x)}{\varphi_t^2(x)}\sum_y \varphi(t,y)r_t(y,x) - f_t(x_t,p_t)\widehat{\varphi}_t(x)\\
&= \sum_{y\neq x} r_t(x,y)\widehat{\varphi}_t(y) + u_t(x,x)\frac{p_t(x)}{\varphi_t(x)} - f_t(x_t,p_t)\widehat{\varphi}_t(x)\\
&= \sum_y r_t(x,y)\widehat{\varphi}_t(y) - f_t(x_t,p_t)\widehat{\varphi}_t(x).
\end{aligned}
$$

$$(50)$$

Eqs. (49) and (50) yield the desired Hopf-Cole system of coupled differential equations, that characterize the solution of the Generalized Schrödinger Bridge in Eq. (34).

**Remark A.3.** *Thus, given the Hopf-Cole transformation, the optimal control can be rewritten as $u_t(y,x) = r_t(y,x)\frac{\varphi(y)}{\varphi(x)}$, and the backward control is given by*

$$
\begin{aligned}
\hat{u}_s(y,x) &:= u_{1-t}(x,y)\frac{p_{1-t}(y)}{p_{1-t}(x)}\\
&= r_{1-t}(x,y)\frac{\varphi_{1-t}(x)}{\varphi_{1-t}(y)}\frac{(\varphi\widehat{\varphi})_{1-t}(y)}{(\varphi\widehat{\varphi})_{1-t}(x)}\\
&= r_{1-t}(x,y)\frac{\widehat{\varphi}_{1-t}(y)}{\widehat{\varphi}_{1-t}(x)}
\end{aligned}
$$

$$(51)$$

*Thus concretely, we have $\hat{u}_s(y,x) = r_{1-t}(x,y)\frac{\widehat{\varphi}_{1-t}(y)}{\widehat{\varphi}_{1-t}(x)}$ with the time-reversed time $s := 1 - t$.*

$\square$

## A.3. Proof of Proposition 3.5

**Proposition A.4.** *Let $(X_t)_{t\in[0,1]}$ be a CTMC process on $\mathcal{X}$, which evolves under the controlled transition rate $u_t(y,x) = e^{Z_t(y,x)}r_t(y,x)$. For convenience, we define*

$$
\begin{aligned}
Y_t(X_t) &:= \log\varphi(t,X_t), \quad Z_t(y,X_t) = Y_t(y) - Y_t(X_t),\\
\widehat{Y}_t(X_t) &:= \log\widehat{\varphi}(t,X_t), \quad \widehat{Z}_t(y,X_t) = \widehat{Y}_t(y) - \widehat{Y}_t(X_t).
\end{aligned}
$$

*Then, the pair $(Y_t, \widehat{Y}_t)$ is expressed through the generator $\mathcal{A}_t^u$ as follows*

$$\mathcal{A}_t^u Y(x) = \sum_y \left(r_t e^{Z_t}\left(Z_t - 1\right)\right)(y,x) + f_t(x,p_t^u)$$

$$(52)$$

$$\mathcal{A}_t^u \widehat{Y}(x) = \sum_y r_t(x,y)e^{\widehat{Z}_t(y,x)} + \left(\widehat{Z}_t r_t e^{Z_t}\right)(y,x) - f_t(x,p_t^u)$$

$$(53)$$

*Moreover, given the backward transition rate $\hat{u}_s(x,y) = e^{\widehat{Z}_s(y,x)} r_s(x,y)$, with $s = 1 - t$*

$$\mathcal{A}_s^{\hat{u}} \widehat{Y}(x) = \sum_y r_s(x,y) e^{\widehat{Z}_s(y,x)} \left( \widehat{Z}_s(y,x) - 1 \right) + f_s(x, p_s^u) \tag{54}$$

$$\mathcal{A}_s^{\hat{u}} Y(x) = \sum_y r_s(x,y) e^{Z_s(y,x)} + Z_s(y,x) r_s(x,y) e^{\widehat{Z}_s(x,y)} - f_s(x, p_s^u)$$

*Proof.* From Proposition 3.4, the log-potentials are defined as

$$Y_t(x) := \log \varphi_t(x), \qquad \hat{Y}_t(x) := \log \widehat{\varphi}_t(x),$$

so that $\log p_t(x) = Y_t(x) + \hat{Y}_t(x)$.

**Lemma A.5.** *The (time-inhomogeneous) generator of the controlled CTMC, acting on a test function $h(t, \cdot)$, is*

$$\mathcal{A}_t^u h(x) = \partial_t h(t, x) + \sum_{y \in \mathcal{X}} \left( h(t, y) - h(t, x) \right) u_t(y, x).$$

*Proof.* Given controlled transition rate $u_t$, note that $X_{t+dt} \sim (I + u_t dt)(\cdot, X_t)$, i.e.

$$p_{t+dt|t}^u(y|x) = \begin{cases} u_t(y, x) dt & \text{when } y \neq x, \\ 1 - \sum_{z \neq x} u_t(z, x) dt & \text{when } y = x. \end{cases} \tag{55}$$

Incorporating probability density, we obtain the following generator for a stochastic function $h(t, \cdot)$, with underlying stochastic process governed by $X_{t+dt} \sim (I + u_t dt)(\cdot, X_t)$:

$$\mathcal{A}_t^u h := \lim_{\Delta t \to 0+} \frac{\mathbb{E}_{y \sim p_{t+\Delta t|t}^u(\cdot|x)} \left[ h(t + \Delta t, y) \right] - h(t, x)}{\Delta t} \tag{56}$$

$$= \frac{1}{dt} \sum_y h(t + dt, y) p_{t+dt|t}^u(y|x) - h(t, X_t) \tag{57}$$

$$= \sum_{y \neq X_t} h(t + dt, y) u_t(y, X_t) - \frac{1}{dt} \sum_{y \neq X_t} u_t(y, X_t) h(t + dt, X_t) + \frac{h(t + dt, X_t) - h(t, X_t)}{dt} \tag{58}$$

$$= \sum_{y \neq X_t} \left( h(t, y) + \frac{\partial h(t, y)}{\partial t} \right) u_t(y, x) - \frac{1}{dt} \sum_{y \neq X_t} \left( h(t, x) + \frac{\partial h(t, x)}{\partial t} dt \right) u_t(y, x) dt + \frac{\partial h}{\partial t}(t, X_t) \tag{59}$$

$$= \sum_{y \neq X_t} \left( h(t, y) - h(t, X_t) \right) u_t(y, x) + \frac{\partial h}{\partial t}(t, X_t) \tag{60}$$

$$= \sum_y \left( h(t, y) - h(t, X_t) \right) u_t(y, x) + \frac{\partial h}{\partial t}(t, X_t) \tag{61}$$

$$\square$$

**Remark A.6.** *Similarly, we can prove that the generator for the backward process obeys:*

$$A_s^u h = \sum_{y \neq x} \left[ h(s - \Delta s, y) - h(s - \Delta s, x) \right] u_s(y|x) - \frac{\partial h}{\partial s}(s, x) \tag{62}$$

*for test function $h$, where recall the backward time coordinate is $s = 1 - t$.*

Recall that from Theorem 3.1, the forward controlled jump rates are

$$u_t(y, x) = r_t(y, x) \, e^{Y_t(y) - Y_t(x)}.$$

**Generator of $\hat{Y}$.** From Eq. (48) and the identity $\partial_t \hat{Y}_t(x) = \widehat{\varphi}_t(x)^{-1} \partial_t \widehat{\varphi}_t(x)$,

$$\partial_t \hat{Y}_t(x) = \frac{1}{\widehat{\varphi}_t(x)} \left( \sum_{y \in \mathcal{X}} r_t(x,y) \widehat{\varphi}_t(y) - f_t(x, p_t) \widehat{\varphi}_t(x) \right)$$

$$= \sum_{y \in \mathcal{X}} r_t(x,y) \exp\big(\hat{Y}_t(y) - \hat{Y}_t(x)\big) - f_t(x, p_t). \tag{63}$$

Therefore,

$$\mathcal{A}_t^u \hat{Y}(x) = \sum_{y \in \mathcal{X}} r_t(x,y) e^{\hat{Y}_t(y) - \hat{Y}_t(x)} - f_t(x, p_t) + \sum_{y \in \mathcal{X}} \big(\hat{Y}_t(y) - \hat{Y}_t(x)\big) \, r_t(y,x) e^{Y_t(y) - Y_t(x)}. \tag{64}$$

**Generator of $Y$.** Similarly, from Eq. (50) and $\partial_t Y_t(x) = \varphi_t(x)^{-1} \partial_t \varphi_t(x)$,

$$\partial_t Y_t(x) = \frac{1}{\varphi_t(x)} \left( -\sum_{y \in \mathcal{X}} r_t(y,x) \varphi_t(y) + f_t(x, p_t) \varphi_t(x) \right)$$

$$= -\sum_{y \in \mathcal{X}} r_t(y,x) \exp\big(Y_t(y) - Y_t(x)\big) + f_t(x, p_t). \tag{65}$$

Plugging Eq. (65) into the generator formula gives the compact form

$$\mathcal{A}_t^u Y(x) = \partial_t Y_t(x) + \sum_{y \in \mathcal{X}} \big(Y_t(y) - Y_t(x)\big) \, r_t(y,x) e^{Y_t(y) - Y_t(x)}$$

$$= f_t(x, p_t) + \sum_{y \in \mathcal{X}} r_t(y,x) e^{Y_t(y) - Y_t(x)} \Big( Y_t(y) - Y_t(x) - 1 \Big). \tag{66}$$

Equations Eq. (64) and Eq. (66) are the claimed generator identities.

**Time-reversed Generators** Similarly, from the expression for the generator of the time-reversed dynamics from Remark A.6 and Eq. (62), along with the optimal control for the backward in time process in Eq. (51), we obtain:

$$\mathcal{A}_s^u Y(x) = \sum_y \big(Y(s,y) - Y(s,x)\big) \, r_s(x,y) e^{\big(\hat{Y}_s(y) - \hat{Y}_s(x)\big)} - \frac{\partial Y}{\partial s}(s,x) \tag{67}$$

where

$$\frac{\partial Y}{\partial s}(s,x) = \frac{1}{\varphi_s(x)} \left( -\sum_y r_s(y,x) \varphi_s(y) + f_s(x, p_t) \varphi_s(x) \right) \tag{68}$$

Similarly, we get for $\widehat{Y}_t$:

$$\mathcal{A}_s^u \hat{Y} = \sum \Big( \hat{Y}(s,y) - \hat{Y}(s,x) \Big) \, r_s(x,y) e^{\big(\hat{Y}_s^\varphi(y) - \hat{Y}_s^\varphi(x)\big)} - \frac{\partial \hat{Y}}{\partial s}(s,x) \tag{69}$$

where

$$\frac{\partial \hat{Y}}{\partial s}(s,x) = \frac{1}{\widehat{\varphi}_s(x)} \left( \sum_y r_s(x,y) \widehat{\varphi}_s(y) - f_s(x, p_t) \varphi_s(x) \right) \tag{70}$$

$\square$

## A.4. Proof of Proposition 3.6

**Proposition A.7.** *Assume the controlled process $(X_t)_{t \in [0,1]}$ on $\mathcal{X}$, sampled with $u_t$ in Eq. 19. Expansion of the generator terms in Eq. (25) gives the forward IPF objective.*

$$\mathcal{L}_{\mathrm{IPF}}^{Z}(\widehat{Z}) := \mathbb{E}_{p^Z}\left[ \int_0^1 \sum_{y \in \mathcal{X}} \left( r_t(X_t, y) e^{\widehat{Z}_t(y, X_t)} + r_t(y, X_t) e^{Z_t(y, X_t)}(-1 + Z_t(y, X_t) + \widehat{Z}_t(y, X_t)) \right) dt \right]. \tag{71}$$

*Similarly for the time reversed dynamics, the corresponding backward IPF objective admits a similar expansion*

$$\mathcal{L}_{\mathrm{IPF}}^{\widehat{Z}}(Z) = \mathbb{E}_{p^{\hat{z}}}\left[ \int_0^1 \sum_{y \in \mathcal{X}} \left( r_t(y, \tilde{X}_t) e^{Z_t(y, \tilde{X}_t)} + r_t(\tilde{X}_t, y) e^{\widehat{Z}_t(y, \tilde{X}_t)}\left( -1 + Z_t(y, \tilde{X}_t) + \widehat{Z}_t(y, \tilde{X}_t) \right) \right) dt \right]. \tag{72}$$

*Proof.* From Lemma A.5, recall that for control function $h_t(x) := \log p_t(x) = Y_t(x) + \hat{Y}_t(x)$, the Dynkin's formula for the controlled process started from $X_0 = x_0$ reads

$$\mathbb{E}\big[g_1(X_1) \,\big|\, X_0 = x_0\big] = g_0(x_0) + \mathbb{E}\Big[ \int_0^1 (\mathcal{A}^u g_t)(X_t)\, dt \,\Big|\, X_0 = x_0 \Big].$$

Rearranging and multiplying by $-1$ gives

$$- \log p_0(x_0) = -\mathbb{E}\big[ \log p_1(X_1) \,|\, X_0 = x_0 \big] + \mathbb{E}\Big[ \int_0^1 (\mathcal{A}^u \log p_t)(X_t)\, dt \,\Big|\, X_0 = x_0 \Big]. \tag{73}$$

In the IPF-style training objective, we replace the true log-potentials $(Y, \hat{Y})$ by their parameterized surrogates $(Y, \widehat{Y})$, yielding an upper bound (tight at the optimum) of the form

$$- \log p_0(x_0) \;\lesssim\; \mathbb{E}\Big[ \int_0^1 \big(\mathcal{A}^u Y_t + \mathcal{A}^u \widehat{Y}\big)(X_t)\, dt \,\Big|\, X_0 = x_0 \Big] := \mathcal{L}_{\mathrm{IPF}}^{Z}(\widehat{Z}) \tag{74}$$

To make Eq. (74) explicit, define the edge-increments

$$Z_t(y, x) := Y_t(y) - Y_t(x), \qquad \hat{Z}_t(y, x) := \widehat{Y}_t(y) - \widehat{Y}_t(x),$$

and recall $u_t(y, x) = r_t(y, x) e^{Z_t(y, x)}$. Using Eq. (64)–Eq. (66), we obtain

$$\mathcal{L}_{\mathrm{IPF}}^{Z}(\widehat{Z}) = \mathbb{E}\left[ \int_0^1 \sum_{y \in \mathcal{X}} \left( r_t(y, x)\, e^{Z_t(y, x)}\big(-1 + Z_t(y, x) + \hat{Z}_t(y, x)\big) \;+\; r_t(x, y)\, e^{\hat{Z}_t(y, x)} \right) dt \,\bigg|\, X_0 = x_0 \right]. \tag{75}$$

This is the forward-pass IPF objective.

**Backward-pass objective.** Applying the same argument to the time-reversed (backward) sampling dynamics started from $X_1 = x_1$, given the expression for the generator of the backward process and the optimal control $\hat{u}_s$, yields the analogous bound

$$- \log p_1(x_1) \;\lesssim\; \mathbb{E}\Big[ \int_0^1 \big(\mathcal{A}_s^u Y + \mathcal{A}_s^u \widehat{Y}\big)(\tilde{X}_s)\, ds \,\Big|\, X_1 = x_1 \Big] := \mathcal{L}_{\mathrm{IPF}}^{\widehat{Z}}(Z),$$

Thus, expanding the generator gives

$$\mathcal{L}_{\mathrm{IPF}}^{\widehat{Z}}(Z) = \mathbb{E}\left[ \int_0^1 \sum_{y \in \mathcal{X}} \left( r_s(x, y)\, e^{\hat{Z}_s(y, x)}\big(-1 + Z_s(y, x) + \hat{Z}_s(y, x)\big) \;+\; r_s(y, x)\, e^{Z_s(y, x)} \right) ds \,\bigg|\, X_1 = x_1 \right]. \tag{76}$$

Minimizing $\mathcal{L}_{\mathrm{IPF}}^{Z}(\widehat{Z})$ and $\mathcal{L}_{\mathrm{IPF}}^{\widehat{Z}}(Z)$ alternately gives the IPF training scheme. $\square$

## B. Connection with Stochastic Optimal Control

Let $p^u$ denote the Markov path measure of a time-inhomogeneous CTMC on $\mathcal{X}$ with controlled jump rates $u_t(\cdot, \cdot)$, and let $p^r$ be the corresponding path measure under the reference rates $r_t(\cdot, \cdot)$. We consider the KL-regularized finite-horizon control problem

$$\min_u \ \mathbb{E}_{X \sim p^u}\left[\int_0^1 f(t, X_t, p_t)\, dt \ + \ g(X_1) \ + \ D_{\mathrm{KL}}(p^u \,\|\, p^r)\right]. \tag{77}$$

This objective is the continuous-time analogue of "soft" control: it trades off running cost and terminal cost against a path-space KL penalty that keeps the controlled dynamics close to the reference CTMC.

**Dynamic programming and one-step expansion.**  Define the value function

$$V(t, x) \ := \ \inf_u \ \mathbb{E}^u\left[\int_t^1 f(\tau, X_\tau, p_\tau)\, d\tau + g(X_1) + D_{\mathrm{KL}}\Big(p^u_{[t,1]} \,\|\, p^r_{[t,1]}\Big) \ \Big| \ X_t = x\right].$$

Applying the dynamic programming principle over a short interval $[t, t + \Delta t]$ yields

$$V(t, x) = \inf_{u_t(\cdot, x)} \left\{\mathbb{E}^u\big[V(t + \Delta t, X_{t+\Delta t}) \mid X_t = x\big] + f(t, x, p_t)\, \Delta t + D_{\mathrm{KL}}\Big(p^u_{[t,t+\Delta t]} \,\|\, p^r_{[t,t+\Delta t]}\Big)\right\}. \tag{78}$$

For a CTMC with rates $u_t(\cdot, x)$, the one-step transition over $\Delta t$ satisfies

$$\mathbb{P}(X_{t+\Delta t} = x \mid X_t = x) = 1 - \sum_{y \neq x} u_t(y, x)\Delta t + o(\Delta t), \qquad \mathbb{P}(X_{t+\Delta t} = y \mid X_t = x) = u_t(y, x)\Delta t + o(\Delta t),$$

and hence

$$\mathbb{E}^u\big[V(t + \Delta t, X_{t+\Delta t}) \mid X_t = x\big] = V(t + \Delta t, x)\Big(1 - \sum_{y \neq x} u_t(y, x)\Delta t\Big) + \sum_{y \neq x} u_t(y, x)\Delta t\, V(t + \Delta t, y) + o(\Delta t). \tag{79}$$

Moreover, the KL divergence between the infinitesimal jump laws induced by $u_t(\cdot, x)$ and $r_t(\cdot, x)$ over $[t, t + \Delta t]$ expands as (up to $o(\Delta t)$)

$$D_{\mathrm{KL}}\Big(p^u_{[t,t+\Delta t]} \,\|\, p^r_{[t,t+\Delta t]}\Big) = \sum_{y \neq x}\Big[u_t(y, x) \log \frac{u_t(y, x)}{r_t(y, x)} - u_t(y, x) + r_t(y, x)\Big]\Delta t + o(\Delta t). \tag{80}$$

**Optimal control and the HJB equation.**  Substituting Eq. (79)–Eq. (80) into Eq. (78), cancelling $V(t + \Delta t, x)$, dividing by $\Delta t$, and letting $\Delta t \to 0$ gives

$$0 = \inf_{u_t(\cdot, x)} \left\{\partial_t V(t, x) + \sum_{y \neq x} u_t(y, x)\big(V(t, y) - V(t, x)\big) + f(t, x, p_t)\right.$$
$$\left. + \sum_{y \neq x}\Big[u_t(y, x) \log \frac{u_t(y, x)}{r_t(y, x)} - u_t(y, x) + r_t(y, x)\Big]\right\}. \tag{81}$$

The minimization in Eq. (81) decouples over $y \neq x$. Differentiating the objective with respect to $u_t(y, x)$ yields the optimal rates

$$u_t^\star(y, x) = r_t(y, x) \exp\big(V(t, x) - V(t, y)\big), \qquad y \neq x. \tag{82}$$

Plugging Eq. (82) back into Eq. (81) and simplifying gives the HJB equation

$$-\partial_t V(t, x) = \sum_{y \neq x} r_t(y, x)\Big(1 - \exp\big(V(t, x) - V(t, y)\big)\Big) + f(t, x, p_t). \tag{83}$$

**Relation to generalized Schrödinger bridges.** Notice that Eq. (83) matches the HJB obtained in Theorem 3.1 of our GSB-on-graphs formulation. The main conceptual difference is in the terminal condition: in the Schrödinger bridge, the endpoint distribution is enforced via hard marginal constraints, whereas in the SOC formulation above the terminal objective is imposed softly through the terminal cost $g(X_1)$ (and, if desired, can be strengthened to approximate hard constraints by choosing $g$ appropriately).

# C. Experimental Details

Across all tasks, we evaluate each method over the *same* finite horizon $T$ and with the *same* rollout budget $B$ particles, using identical endpoint marginals $(p_0, p_1)$ and the same evaluation code for metrics. For baselines, we allocate an equal tuning budget (grid over $\leq M$ settings) and report the best configuration on a held-out validation objective.

## C.1. Assignment

**Balanced assignment as graph transport.** We demonstrate the effectiveness of GSBoG on balanced assignment problems between $n$ supply nodes $\{A_i\}_{i=1}^n$ and $n$ demand nodes $\{B_j\}_{j=1}^n$, with *source* and *target* marginals $p_0, p_1 \in \Delta^n$ (with $n \in \{6, 8, 10, 20\}$), where $p_0(A_i) = (p_0)_i$ and $p_1(B_j) = (p_1)_j$.

Given costs $C \in \mathbb{R}^{n \times n}$, the *oracle* transport plan solves the static linear program

$$\pi^\star \in \arg\min_{\pi \geq 0} \langle C, \pi \rangle \quad \text{s.t.} \quad \pi \mathbf{1} = p_0, \ \ \pi^\top \mathbf{1} = p_1. \tag{84}$$

**Graph encoding of pairwise costs (using $x$ as current node and $y$ as next node).** To express pairwise assignment costs via *state costs* on a graph, we introduce intermediate nodes $E_{ij}$ for every pair $(i, j)$ and build a directed graph $\mathcal{G} = (V, E)$ with

$$V = \{A_i\}_{i=1}^n \ \cup \ \{E_{ij}\}_{i,j=1}^n \ \cup \ \{B_j\}_{j=1}^n.$$

Edges are

$$(x, y) \in E \ \Leftrightarrow \ \big(x = A_i, \ y = E_{ij}\big) \text{ or } \big(x = E_{ij}, \ y = B_j\big), \qquad \forall i, j,$$

(and optionally self-loops $(x, x)$ for numerical stability). We define the running *state cost* $f : V \to \mathbb{R}_+$ by

$$f(x) := \begin{cases} C_{ij}, & x = E_{ij}, \\ 0, & x \in \{A_i\}_{i=1}^n \cup \{B_j\}_{j=1}^n. \end{cases} \tag{85}$$

The endpoint distributions over $V$ are supported on the supply/demand partitions:

$$p_0(x) = \begin{cases} (p_0)_i, & x = A_i, \\ 0, & \text{otherwise}, \end{cases} \qquad p_1(x) = \begin{cases} (p_1)_j, & x = B_j, \\ 0, & \text{otherwise}. \end{cases} \tag{86}$$

Throughout, we use $x$ to denote the *current* node and $y$ the *next* node along a transition (e.g., in discrete time $X_{t+1} = y$ given $X_t = x$, or in continuous time a rate $u_t(y, x)$ for jumps $x \to y$).

**Decoding a soft assignment from learned dynamics.** Let $(X_t)_{t=0}^T$ denote a rollout under the learned dynamics. We decode a soft plan $\hat{\pi} \in \mathbb{R}_+^{n \times n}$ from the expected edge flux on the supply-to-intermediate edges:

$$\hat{\pi}_{ij} \ \propto \ \mathbb{E}\left[ \sum_{t=0}^{T-1} \mathbf{1}\{X_t = A_i, \ X_{t+1} = E_{ij}\} \right], \tag{87}$$

followed by a normalization chosen so that $\hat{\pi} \mathbf{1} = p_0$ and $\hat{\pi}^\top \mathbf{1} = p_1$ (up to numerical tolerance). We then compare $\hat{\pi}$ against the oracle plan $\pi^\star$.

**Instance generation.** For each $n$, we generate synthetic instances by sampling (i) marginals from a Dirichlet distribution (e.g., $p_0 \sim \text{Dir}(\alpha \mathbf{1})$, $p_1 \sim \text{Dir}(\alpha \mathbf{1})$ with $\alpha = 1$), and (ii) a structured cost matrix via 2D embeddings: sample points $u_i, v_j \in [0, 1]^2$ uniformly and set

$$C_{ij} := \|u_i - v_j\|_2. \tag{88}$$

In practice, any other deterministic cost construction can be used instead of equation 88.

*Table 5.* Assignment task metrics

| $n$ | Optimal Cost | Assigned Cost | Mass on opt. selection | Row Entropy | Accuracy |
|---|---|---|---|---|---|
| 6 | 160 | 160 | 97.3% | 0.07 | 100% |
| 8 | 205 | 205 | 94.2% | 0.13 | 100% |
| 10 | 239 | 239 | 87.6% | 0.16 | 100% |
| 20 | 538 | 546 | 85.0% | 0.22 | 90% |

**Baselines.** We compare against (i) the exact min-cost flow / OT solution of Eq. (84), used as ground truth $\pi^\star$.

**Metrics.** For each method returning a soft plan $\tilde{\pi}$, we report

$$\textbf{Cost: } \langle C, \tilde{\pi} \rangle, \qquad \textbf{Cost gap: } \frac{\langle C, \tilde{\pi} \rangle - \langle C, \pi^\star \rangle}{\langle C, \pi^\star \rangle}, \tag{89}$$

$$\textbf{Marginal error (TV): } \mathrm{TV}(\tilde{\pi}\mathbf{1}, p_0) + \mathrm{TV}(\tilde{\pi}^\top \mathbf{1}, p_1), \quad \mathrm{TV}(p,q) := \tfrac{1}{2}\|p - q\|_1, \tag{90}$$

$$\textbf{Mean row entropy: } \frac{1}{n}\sum_{i=1}^{n} H(\tilde{\pi}_{i:}), \quad H(r) := -\sum_{j=1}^{n} r_j \log(r_j + \epsilon), \tag{91}$$

$$\textbf{Mass on optimal edges: } \sum_{(i,j)\in\mathrm{supp}(\pi^\star)} \tilde{\pi}_{ij}, \qquad \textbf{Edge overlap (Accuracy): } \frac{|\{(i, \hat{\sigma}(i))\} \cap \mathrm{supp}(\pi^\star)|}{n}. \tag{92}$$

### C.2. Supply Chain

To study the capacity of our model in navigating complex and sparse dynamics on a graph, we model a real-world supply-chain network (Kovács, 2015), as a directed graph $\mathcal{G} = (\mathcal{X}, \mathcal{E})$, where each node $x \in \mathcal{X}$ represents a facility/location (e.g., supplier, depot, port, retailer), and each directed edge $(x, y) \in \mathcal{E}$ represents an admissible shipment arc. As the scale of modern decision-making increases, there is an emerging need for effective architectures to handle high-dimensional problems (Saravanos et al., 2025). The graph is illustrated in Figure 7. We define the sparse CTMC reference generator $r_{\mathrm{ref}} \in \mathbb{R}^{|\mathcal{X}| \times |\mathcal{X}|}$ supported on $\mathcal{E}$: for $(x, y) \in \mathcal{E}$, $r(y, x) := (r_{\mathrm{ref}})_{yx} \geq 0$ denotes the base jump rate, and the diagonal is set by $(r_{\mathrm{ref}})_{xx} = -\sum_{y\neq x} r(y, x)$ so that columns sum to zero. This reference process captures the *topology*, i.e., which moves are possible and a nominal routing bias (e.g., proportional to edge capacities or inverse travel time), without enforcing the boundary marginals. Table 6 summarizes the properties of the considered graph.

The boundary marginals $p_0, p_1 \in \Delta^{|\mathcal{X}|-1}$, where $\Delta^{|\mathcal{X}|-1}$ is the probability simplex. Mass on source nodes $\mathcal{S}$ and demand nodes $\mathcal{D}$, uniform or volume-weighted, we solve a finite-horizon transport problem over $T = 100$ steps with uniform discretization, learning a topology-respecting stochastic routing policy whose rollouts start from $p_0$ and reach $p_1$, while optimizing operational objectives *during transit*.

To reflect operational constraints beyond shortest-path routing, we include a state and distribution dependent running cost $f_t(x, p_t)$ that penalizes undesirable locations, such as congested hubs, encouraging trajectories to route around high-cost/high-occupancy nodes while still matching $(p_0, p_1)$ within the prescribed horizon. Concretely, we penalize interaction-driven congestion by defining node occupancy

$$\mathrm{occ}_t(x) = \sum_{b=1}^{B} \mathbf{1}\{X_t^{(b)} = x\}, \quad t = 1, \dots, T, x \in \mathcal{X}$$

excluding the source and target endpoint nodes, and charge each particle the exposure. This setting highlights where GSBoG is strongest: it (i) learns an *executable policy over paths* rather than a static coupling, (ii) operates directly on the given sparse topology so every transition is feasible, and (iii) natively supports state-/path-dependent costs such as congestion induced by collective occupancy. By contrast, conventional graph OT and network-flow baselines typically output a static plan, which is not optimized for intermediate behavior and can concentrate traffic through hubs.

*Table 6.* Road-flow graph statistics (DIMACS min-cost flow instance).

| Statistic | Value |
|---|---|
| Nodes $N$ | 9559 |
| Sparsity | 99.97 % |
| Density (unique edges) | $3 \times 10^{-4}$ |
| Mean out-degree / in-degree | 3.105 / 3.105 |
| Out-degree min / median / max | 1 / 3 / 6 |
| In-degree min / median / max | 1 / 3 / 6 |
| Arc cost min / median / mean / max | 1.0 / 250 / 293.3 / 4537 |
| Arc capacity min / median / mean / max | 120 / 480 / 459.6 / 480 |

*Table 7.* Supply and demand nodes (net imbalances) for the road-flow instance.

| Type | Node ids (amount) |
|---|---|
| Supply (+) | 6923 (+1920), 5780 (+1440), 8243 (+1080), 5250 (+960), 7093 (+480), 5570 (+480) |
| Demand (–) | 4455 (–1920), 7404 (–1440), 2490 (–1440), 1889 (–600), 5218 (–480), 2105 (–480) |

*Table 8.* Capacity distribution over directed arcs.

| Capacity | # arcs | Share |
|---|---|---|
| 120 | 476 | 1.60% |
| 240 | 674 | 2.26% |
| 360 | 2288 | 7.69% |
| 480 | 26330 | 88.45% |

**Baselines** We present the baselines used for the supply chain network experimental setup. All methods are evaluated using the same horizon $T = 100$ with uniform discretization, and rollout budget $B = 5,000$.

**Schrödinger Bridge on graphs** Let $\mathcal{G} = (V, E)$ be a weighted graph and let $\rho_0, \rho_1$ be endpoint distributions. Chow et al. (2022) formulate the *dynamical Schrödinger bridge on graphs* as

$$\min_{\rho(\cdot), b(\cdot)} \int_0^1 \frac{1}{2} \langle b(t), b(t) \rangle_{\rho(t)} \, dt \quad \text{s.t.} \quad \begin{cases} \dot{\rho}(t) + \operatorname{div}_{\mathcal{G}}\Big( \rho(t)\big( b(t) - \beta \nabla_{\mathcal{G}} \log \rho(t) \big) \Big) = 0, \\ \rho(0) = \rho_0, \qquad \rho(1) = \rho_1, \end{cases} \tag{93}$$

where $\beta > 0$ controls the entropic/noise level and $\nabla_{\mathcal{G}}$, $\operatorname{div}_{\mathcal{G}}$ are graph gradient/divergence operators; the inner product $\langle \cdot, \cdot \rangle_p$ encodes a discrete $W_2$-type geometry on $\Delta^{N-1}$.

*Hamiltonian flow on the simplex.* Introducing a node potential (co-state) $\phi(t) \in \mathbb{R}^N$, the first-order optimality conditions of Eq. (93) can be written as a two-point boundary-value Hamiltonian system:

$$\dot{\rho}(t) = \nabla_\phi \, \mathcal{H}\big( \rho(t), \phi(t) \big), \qquad \dot{\phi}(t) = -\nabla_p \, \mathcal{H}\big( \rho(t), \phi(t) \big), \qquad \rho(0) = \rho_0, \ \rho(1) = \rho_1, \tag{94}$$

for a Hamiltonian $\mathcal{H}(p, \phi)$ induced by the discrete kinetic energy and the entropic/noise term. Thus, the bridge is obtained by solving a symplectic BVP whose marginals $\rho(t)$ evolve as a Hamiltonian flow on the probability simplex $\Delta^{N-1}$.

*Computation.* Numerically, Eq. (94) is a coupled ODE system in $\mathbb{R}^{2N}$ with one pair $(\rho_i(t), \phi_i(t))$ per node, coupled through the graph operators, yielding a large boundary-value problem whose dimension scales with $N$.

**Attraction flow.** We include a graph-native *attraction-flow* baseline that iteratively transports mass toward the target marginal using a Laplacian potential. This baseline constructs a $T$-hop transport process by repeatedly pushing mass along edges in the direction of a target-attracting potential. At each hop $t = 0, \dots, T - 1$, a node potential $\phi_t$ is computed from the discrepancy between the current marginal $\rho_t$ and the target $\rho_1$ (via a Laplacian/Poisson solve on the graph). Mass is then routed from a node $x$ to a neighbor $y$ in proportion to the downhill potential drop $\phi_t(x) - \phi_t(y)$, producing a directed flow

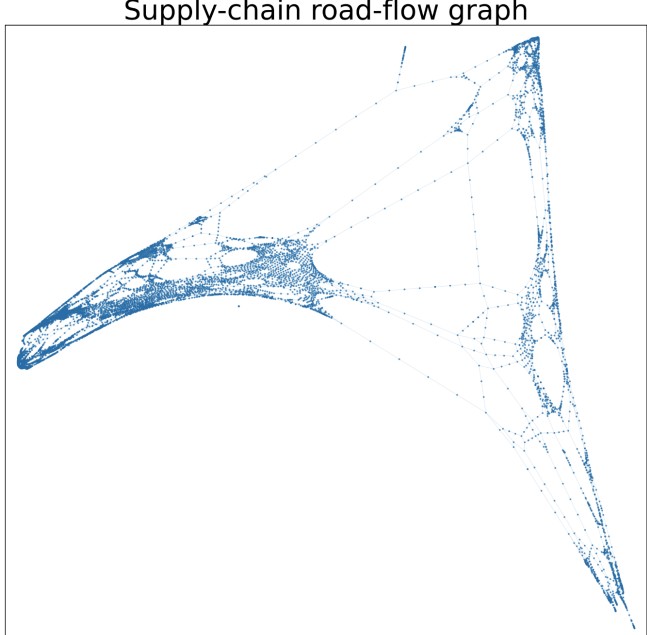

*Figure 7.* Graph of our supply demand setting.

$\rho_t(x \to y)$. To obtain an *actionable* policy for rollout-based evaluation, we apply a *Markovian embedding* by normalizing the outgoing flow, $P_t(y \mid x) \propto \rho_t(x \to y)$ (with optional self-loop/fallback mass when needed), and simulate trajectories via $x_{t+1} \sim P_t(\cdot \mid x_t)$. Embedding parameters are selected to minimize terminal mismatch (TV) under the same horizon and rollout budget used for GSBoG, ensuring a fair comparison.

**Time-expanded dynamic $\mathcal{W}_1$ (min-cost flow / discrete Benamou–Brenier).** We include an *exact* dynamic $\mathcal{W}_1$ baseline obtained by solving a single *time-expanded* min-cost flow problem over a horizon of $T$ steps. Given a directed graph $\mathcal{G} = (\mathcal{X}, \mathcal{E})$ with per-edge capacities $\mathrm{cap}(x \to y)$ and costs $c(x \to y)$, we build a layered network with nodes $(t, x)$ for $t = 0, \ldots, T$ and $x \in \mathcal{X}$. For each original edge $(x \to y) \in \mathcal{E}$, we add a transport arc $(t, x) \to (t+1, y)$ for every $t = 0, \ldots, T-1$, inheriting the same capacity and cost. To allow *waiting* (so the per-step policy is always well-defined), we also add holdover arcs $(t, x) \to (t+1, x)$ with cost 0 and capacity $M$.

Let $\rho_0, \rho_1 \in \Delta^{|\mathcal{X}|-1}$ denote endpoint marginals, and let $M$ be the total transported mass (used to scale probabilities into flow units). We solve for nonnegative per-step arc flows $q_t(x \to y)$ and holdover flows $q_t(x \to x)$:

$$\min_{q \geq 0} \sum_{t=0}^{T-1} \left( \sum_{(x \to y) \in \mathcal{E}} c(x \to y) \, q_t(x \to y) \right) \quad \text{s.t.} \qquad (95)$$

$$\begin{cases} \sum_{y:(x \to y) \in \mathcal{E}} q_0(x \to y) + q_0(x \to x) = M \, p_0(x), & \forall x, \\ \sum_{y:(x \to y) \in \mathcal{E}} q_t(x \to y) + q_t(x \to x) = \sum_{y:(y \to x) \in \mathcal{E}} q_{t-1}(y \to x) + q_{t-1}(x \to x), & \forall x, \ t = 1, \ldots, T-1, \\ \sum_{y:(y \to x) \in \mathcal{E}} q_{T-1}(y \to x) + q_{T-1}(x \to x) = M \, p_1(x), & \forall x, \\ 0 \leq q_t(x \to y) \leq \mathrm{cap}(x \to y), & \forall (x \to y) \in \mathcal{E}, \ \forall t, \\ 0 \leq q_t(x \to x) \leq M, & \forall x, \ \forall t. \end{cases} \qquad (96)$$

This yields a time-indexed feasible flow $\{q_t\}_{t=0}^{T-1}$ whose induced node marginals match $p_0$ at $t = 0$ and $p_1$ at $t = T$. To obtain an actionable policy, we again use a *Markovian embedding* $P_t(y \mid x) \propto \rho_t(x \to y)$ and roll out trajectories with $x_{t+1} \sim P_t(\cdot \mid x_t)$, tuning any embedding smoothing/fallback mass to minimize terminal mismatch (TV) under the same rollout budget as GSBoG.

*Remark (capacity metrics and robustness).* The min-cost flow enforces capacities on the *fractional* plan $\rho_t$, but our reported

capacity statistics are computed from *sampled rollouts* of the embedded policy. Notably, although the underlying flux can be capacity-feasible in the time-expanded formulation, while its stochastic execution concentrates mass on a small set of bottleneck edges In particular, the flow solution saturates many edges ($\rho_t(x \to y) \approx \text{cap}(x \to y)$), stochastic sampling yields transient edge loads exceeding capacity, so rollout-based $\max(\text{flow}/\text{cap})$ may be $> 1$ despite a feasible underlying flow. In contrast, GSBoG's congestion-aware objective typically avoids widespread saturation and leaves more slack, leading to policies that are empirically more *robust* to rollout variability while still matching the endpoints.

**Metrics.** Let $\{X_t^{(b)}\}_{t=0}^T$ for $b = 1, \ldots, B$ denote $B$ rollout trajectories over a common horizon of $T$ discrete steps. Let $n_t(x) := \sum_{b=1}^B \mathbf{1}\{X_t^{(b)} = x\}$ be the number of particles at node $x$ at step $t$ (so $\sum_x n_t(x) = B$), and define the empirical marginal

$$\hat{p}_t(x) := \frac{1}{B} \sum_{b=1}^B \mathbf{1}\{X_t^{(b)} = x\}$$

- **Terminal mismatch.** We form the empirical terminal distribution $\hat{p}_T$ and report

$$\text{TV}(\hat{p}_T, p_1) = \frac{1}{2} \sum_{x \in \mathcal{X}} \left| \hat{p}_T(x) - p_1(x) \right|.$$

- **Occupancy and congestion.** We use the occupancy $\text{occ}_t(x)$, and report peak occupancy $\max_{t,x} \text{occ}_t(x)$ and mean congestion over the top-100 most occupied nodes,

$$\frac{1}{T |I_{100}|} \sum_{t=1}^T \sum_{x \in I_{100}} \text{occ}_t(x), \quad I_{100} := \text{Top100}\Big(\{S(x)\}_{x \in \mathcal{X}}\Big), \;\; S(x) := \sum_{t=1}^T \text{occ}_t(x),$$

  excluding the endpoint nodes (source and target). Note, that since all methods use the same $B$, these quantities are directly comparable among all methods.

- **Capacity feasibility.** Given per-step edge capacities $\text{cap}(e) > 0$, we estimate the per-step edge flow from rollouts as

$$\widehat{\text{flow}}_t(e) := \sum_{b=1}^B \mathbf{1}\{(X_t^{(b)}, X_{t+1}^{(b)}) = e\},$$

  We report violations using

$$V_{\max} := \max_{t,e} \frac{\widehat{\text{flow}}_t(e)}{\text{cap}(e)},$$

  A method is *capacity-feasible* if $V_{\max} \le 1$

*Remark* We do not explicitly optimize edge-capacity constraints; however, congestion-aware policies often reduce violations implicitly by spreading flow over alternative routes rather than collapsing onto bottlenecks.

**GSBoG implementation and hyperparameters** We implement GSBoG in PyTorch using sparse graph operations over the directed edge list. At each iteration, we simulate batches of $B$ particles for $T$ discrete time steps (uniform step size $\Delta t$) under the current forward/backward rate parametrizations induced by the learned potentials, and estimate intermediate occupancies $\hat{p}_t$ by empirical counts. Potentials $Y^\theta(t, x)$ and $\widehat{Y}^\phi(t, x)$ are modeled with a shared node embedding and a small MLP conditioned on (normalized) time $t$, producing scalar outputs per node; controlled rates are formed locally on edges as $u_t(y, x) = r_t(y, x) \exp(Y^\theta(t, y) - Y^\theta(t, x))$ (and analogously for $\widehat{u}$), ensuring graph-feasible transitions without dense solvers. We train by alternating forward/backward updates with the gIPF endpoint likelihood objective and the TD regularizer with $\lambda_{\text{TD}} \in \{0, 0.05, 0.1, 0.2, 0.3, 0.5\}$, using AdamW with learning rate $\eta = 5 \cdot 10^{-5}$. The results in Table 2 were obtained with $\lambda_{\text{TD}} = 0.2$.

**Ablation study on $\lambda_{\text{TD}}$**    Figure 8 studies the effect of the tradeoff parameter $\lambda_{\text{TD}}$ on terminal accuracy and intermediate crowding. As $\lambda_{\text{TD}}$ increases, the peak occupancy decreases monotonically (from $\sim 320$ to $\sim 240$), indicating that the induced dynamics spread mass more evenly and avoid bottlenecks. This reduction in crowding comes at the cost of worse terminal matching: the total variation (TV) between the final distribution and the target increases (from $\sim 2.3 \times 10^{-2}$ to $\sim 3.7 \times 10^{-2}$). The curves exhibit a clear knee around $\lambda_{\text{TD}} \approx 0.2$, where congestion drops sharply for only a moderate increase in TV; beyond this point, further congestion decrease is marginal while terminal mismatch continues to grow. Overall, $\lambda_{\text{TD}}$ controls an accuracy–feasibility tradeoff, with $\lambda_{\text{TD}} \approx 0.2$ offering a favorable balance in this experiment.

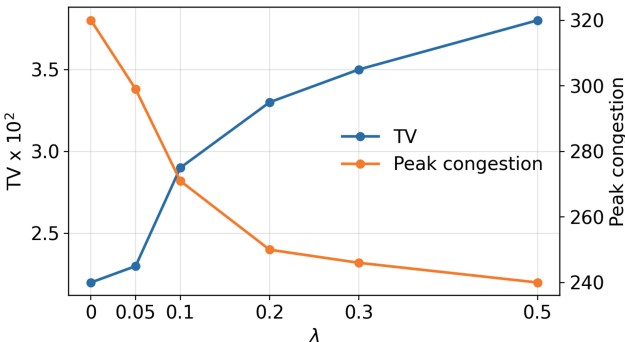

*Figure 8.* Ablation on $\lambda_{\text{TD}}$ showning decrease of the peak congestion, and a slight increase in total variation.

### C.3. Discretized Molecular Dynamics

**Chignolin**    Chignolin (CLN025 variant) is a widely used benchmark for molecular simulation because it is a minimal, 10-residue peptide (166 atoms) that nonetheless preserves the essential features of protein folding (Lindorff-Larsen et al., 2011). Specifically, it exhibits well-separated unfolded and folded metastable configurations connected by rare, activated transitions, making it a useful testbed for modeling methods that aim to resolve thermodynamics and kinetics of protein folding while remaining computationally tractable.

**Reference simulation and MSM construction**    The reference molecular dynamics (MD) simulation trajectories at $T_{\text{sim}} = 300$ K for chignolin were obtained from Iida (2022), and conformational states were discretized into MSM using PyEMMA (Scherer et al., 2015). As input features, we used the backbone torsion angles ($\phi$ and $\psi$) encoded as their sine and cosine components. We used k-means clustering to partition the torsion features into $|\mathcal{X}| = 500$ microstate clusters. For computational efficiency, clustering was performed on a subsampled set of frames with a stride of 10. Then, an MSM was estimated from discretized trajectories at a fixed lag time of 10 steps (in units of the sampling interval, corresponding to 1 ns simulation time). Transition counts between microstates separated by this lag time were accumulated across trajectories and normalized to obtain the microstate-to-microstate transition probability matrix. From this, the stationary distribution $\pi_i$ over microstates $i \in \mathcal{X}$ was computed, and microstate free energies (in units of $k_{\text{B}}T$) were obtained as $F_i / k_{\text{B}}T_{\text{sim}} = -\log \pi_i$.

To define folded/unfolded basins of microstates, we start from computing the aligned root-mean-squared deviation (RMSD) of the ten $\text{C}_\alpha$ atoms between each frame $\tilde{X}_i \in \mathbb{R}^{10 \times 3}$ and the native folded structure (PDB 5AWL) $\tilde{Y} \in \mathbb{R}^{10 \times 3}$. The aligned RMSD is defined as

$$\text{RMSD}(X, Y) := \sqrt{\frac{1}{N} \min_{R \in \text{SE}(3)} \|X \circ R - Y\|_F^2} \tag{97}$$

for 3-D point clouds $X, Y \in \mathbb{R}^{N \times 3}$, where $R = (Q, t) \in \text{SE}(3)$ with $Q \in \text{SO}(3)$, $t \in \mathbb{R}^3$ acts on $X$ as $X \circ R := XQ + \mathbf{1}t^\top$. With an initial RMSD threshold of $d_{\text{folded}} = 0.09$ nm and $d_{\text{unfolded}} = 0.5$ nm, for each microstate, we compute folded/unfolded probabilities as a fraction of frames in that microstate having RMSD smaller than $d_{\text{folded}}$ and larger than $d_{\text{unfolded}}$, respectively. Finally, we assign microstates with folded probability higher than 0.9 to the folded basin $\mathcal{F}$, and unfolded probability higher than 0.8 to the unfolded basin $\mathcal{U}$. Endpoint distributions $\mu$ and $\nu$ are defined by microstate probabilities restricted to unfolded and folded basins: $\mu(i) = \pi_i \mathbf{1}[i \in \mathcal{U}] / \sum_{k \in \mathcal{U}} \pi_k$ and $\nu(i) = \pi_i \mathbf{1}[i \in \mathcal{F}] / \sum_{k \in \mathcal{F}} \pi_k$. Additionally, we compute a mean $\text{C}_\alpha$ RMSD per microstate $r_i$ by averaging the $\text{C}_\alpha$ RMSD of the frames in the microstate.

Prior to defining the CTMC reference dynamics, we prune long-range edges in the reference MSM graph that make large

geometric jumps. Given the MSM transition matrix $T^{\text{MSM}}$ at the chosen lag time, we remove transitions whose RMSD jump exceeds a threshold $\delta = 0.1$ nm, i.e., we set $T_{ij}^{\text{MSM}} \leftarrow 0$ when $|d_i - d_j| > \delta$. To avoid disconnecting the graph, for any state $i$ whose outgoing (or incoming) edges would all be removed by the previous criterion, we retain the single edge with the smallest RMSD difference. The resulting sparse transition matrix $T_{ij}^{\text{MSM,pruned}}$ is then converted into a sparse CTMC generator by assigning off-diagonal rates proportional to the surviving transition probabilities and enforcing the column sum to be zero: $(Q_{\text{ref}})_{ij} = \frac{1}{\tau} T_{ij}^{\text{MSM,pruned}}$ for $i \neq j$ and $(Q_{\text{ref}})_{ii} = -\sum_{j \neq i} (Q_{\text{ref}})_{ij}$, where $\tau = 0.1$ is a hyperparameter that relates CTMC timescale to the MSM lag time. This construction is equivalent to a first-order mapping $T^{\text{MSM}} = e^{\tau Q} \approx I + \tau Q$, while preserving the pruned MSM topology and preventing discrete trajectories from making large structural transitions.

**Baselines**   We compare GSBoG against two graph-native transport baselines and a kinetics-inspired biasing baseline. First, we include the dynamical Schrödinger bridge formulation on finite graphs of Chow et al. (2022) (GraphSB), which solves a simplex-valued boundary-value system to produce a time-inhomogeneous Markov process matching the prescribed endpoints. Second, we include an *attraction-flow* baseline that induces a $T$-step Markov process by turning a static edge flow / potential-driven routing rule into a row-stochastic transition kernel and rolling it out for $T$ steps.

In addition to the SB-based baselines, we include a committor-guided biased dynamics baseline. The forward committor for microstate $i$ is defined as $q(i) := \mathbb{P}_i(\tau_{\mathcal{F}} < \tau_{\mathcal{U}})$, i.e., the probability that a trajectory initiated at $i$ reaches $\mathcal{F}$ before $\mathcal{U}$, with boundary conditions $q(i) = 0$ for $i \in \mathcal{U}$ and $q(i) = 1$ for $i \in \mathcal{F}$ (Vanden-Eijnden et al., 2010). For interior states $\mathcal{I} = \mathcal{X} \setminus (\mathcal{U} \cup \mathcal{F})$, $q$ satisfies the Dirichlet problem $q(i) = \sum_j T_{ij}^{\text{MSM}} q(j)$, $i \in \mathcal{I}$. To bias sampling toward folding, we apply a Doob-$h$ transform (Doob, 1957) that reweights transitions toward larger committor values: $T_{ij}^{(h)} = T_{ij}^{\text{MSM}} h(j) / \sum_k T_{ik}^{\text{MSM}} h(k)$, and we use trajectories sampled from this biased Markov chain.

**Metrics**   We report the maximum state free energy attained during rollout (free energy barrier) and the fraction of trajectories that correctly terminate in the folded basin $\mathcal{F}$ (fold rate) in Tab. 4. More specifically, let $F(x)/k_B T_{\text{sim}} = -\log \pi_x$ be the reference MSM microstate free energy. For a rollout trajectory $(X_t)_{t=0}^T$, we define its barrier as

$$\Delta F(X_{0:T}) := \max_{0 \leq t \leq T} F(X_t) - \min_{x \in \text{supp}(\mu) \cup \text{supp}(\nu)} F(x),$$

Thus, lower $\Delta F_{\text{eff}}$ indicates the induced dynamics reaches the folded basin while avoiding high-$F$ intermediates under the *reference* MSM energies. We also report the $C_\alpha$ RMSD averaged across independent trajectories at each rollout time in Fig. 5.

**GSBoG implementation and hyperparameters**   We run GSBoG directly on the MSM microstate graph, with nodes given by MSM microstates and edges given by the nonzero support of the MSM transition matrix. Training follows our alternating forward/backward IPF scheme with an additional TD-style penalty to incorporate nontrivial $f$ when used: we repeatedly (i) roll out batches of trajectories under the current forward (resp. backward) controlled dynamics, (ii) form the corresponding IPF losses from the discrete-time generator/Dynkin identities, and (iii) update $(Y, \widehat{Y})$. We use a fixed horizon of $T = 200$ steps, $B = 1024$ rollouts, 100 IPF iterations, the value for the $\lambda_{\text{TD}}$ was set to 0.2, AdamW optimization with learning rate $\eta = 10^{-4}$, and an MLP parameterization.

## D. Extended Related Works

**Schrödinger Bridge**   Recently, generative modeling has seen a wave of principled methods rooted in Optimal Transport (Villani et al., 2009). A particularly influential formulation is the Schrödinger Bridge (SB; Schrödinger (1931)). SB became especially prominent in generative modeling after work introduced an Iterative Proportional Fitting (IPF) training procedure—essentially a continuous-state analogue of Sinkhorn—for solving the dynamic SB problem (De Bortoli et al., 2021; Vargas et al., 2021). Conceptually, SB extends standard diffusion modeling beyond fixed endpoint choices by enabling transport between arbitrary distributions $\pi_0$ and $\pi_1$ via fully nonlinear stochastic dynamics, and it identifies the (unique) path measure that minimizes a kinetic-energy–type objective. The Schrödinger Bridge problem (Schrödinger, 1931) in the path measure sense seeks the optimal measure $\mathbb{P}^\star$ that minimizes the following minimization

$$\min_{\mathbb{P}} \text{KL}(\mathbb{P}|\mathbb{Q}), \quad \mathbb{P}_0 = \pi_0, \ \mathbb{P}_1 = \pi_1 \tag{98}$$

where $\mathbb{Q}$ is a Markovian reference measure. Hence the solution of the dynamic SB $\mathbb{P}^\star$ is considered to be the closest path measure to $\mathbb{Q}$. Another formulation of the dynamic SB crucially emerges by applying the Girsanov theorem framing the

problem as a Stochastic Optimal Control (SOC) Problem (Chen et al., 2016; 2021).

$$\min_{u_t,p_t} \int_0^1 \mathbb{E}_{p_t}[\|u_t\|^2]dt \quad \text{s.t.} \quad \frac{\partial p_t}{\partial t} = -\nabla \cdot (u_t p_t) + \frac{\sigma^2}{2}\Delta p_t, \quad \text{and} \quad p_0 = \pi_0, \quad p_1 = \pi_1 \tag{99}$$

Recall, the connection between our GSB objective and the corresponding SOC problem presented in Sec. B. Finally, note that the static SB is equivalent to the entropy regularized OT formulation (Pavon et al., 2021; Nutz, 2021; Cuturi, 2013).

$$\min_{\pi \in \Pi(\pi_0,\pi_1)} \int_{\mathbb{R}^d \times \mathbb{R}^d} \|x_0 - x_1\|^2 d\pi(x_0, x_1) + \epsilon \mathrm{KL}(\pi|\pi_0 \otimes \pi_1) \tag{100}$$

This regularization term enabled efficient solution through the Sinkhorn algorithm and has presented numerous benefits, such as smoothness, and other statistical properties (Ghosal et al., 2022; Léger, 2021; Peyré & Cuturi, 2019).

**Iterative Proportional Fitting (IPF) for continuous Schrödinger Bridges.** A standard way to view the Schrödinger Bridge (SB) is as a *KL projection on path space*: among all path measures whose endpoints match $(\pi_0, \pi_1)$, SB selects the one closest (in relative entropy) to a reference Markov process. This variational structure naturally yields *Iterative Proportional Fitting* (IPF), i.e., alternating KL projections onto the sets of path measures that satisfy one endpoint constraint at a time. In finite spaces, this specialization recovers the classical Sinkhorn/iterative scaling procedure for the Schrödinger problem and entropic OT, equivalently iterating the Schrödinger system via multiplicative reweighting of forward/backward potentials. In continuous-state diffusions, IPF can be interpreted as alternately refining the forward and backward dynamics (or their associated potentials) so that the induced time-marginals progressively enforce the endpoint constraints, connecting SB computation to stochastic control and PDE/FBSDE representations. Recent generative-modeling work focuses on making these IPF updates practical in high dimension by replacing exact potential/drift updates with learned approximations. Diffusion Schrödinger Bridge (DSB; (De Bortoli et al., 2021)) explicitly frames training as an approximation to IPF, where each iteration fits forward/backward score/drift models so that terminal marginals better match the prescribed endpoints. Complementary approaches reinterpret the alternating updates through likelihood or divergence-based objectives: Vargas (2021) propose a maximum-likelihood route to SB learning that can be cast as an alternating procedure enforcing endpoint consistency, while SB-FBSDE (Chen et al., 2021) leverages a nonlinear Feynman–Kac/FBSDE formulation to obtain tractable training losses without gridding the state space. Deep Generalized Schrödinger Bridge (DeepGSB; (Liu et al., 2022b)) further emphasizes that alternating minimization between parametrized forward/backward path measures corresponds to iterative KL projection—hence IPF—and extends this template to mean-field interactions, yielding a TD-learning-like computational structure. Beyond generative modeling, related sample-based and data-driven variants adapt Fortet/Sinkhorn-style iterations when only samples from the endpoint distributions are available (Pavon et al., 2021).

**Iterative Markovian Fitting (IMF) for continuous Schrödinger Bridges.** More recently, in continuous spaces substantial progress has been made on scalable Schrödinger bridge (SB) solvers. Peluchetti (2023) propose a *Markovian projection* viewpoint for SBs, providing a principled route to Markovian dynamics that reproduce target marginals. Building on this interpretation, Diffusion Schrödinger Bridge Matching (DSBM; (Shi et al., 2023)) uses Iterative Markovian Fitting (IMF) to approximate the SB solution. Relatedly, De Bortoli et al. (2023) study flow- and bridge-matching processes and introduce modifications aimed at preserving coupling information, demonstrating efficient learning on mixtures of image-translation tasks. $SF^2\text{-}M$ (Tong et al., 2023) provides a simulation-free objective for inferring stochastic dynamics and is effective in practice for SB problems, while GSBM (Liu et al., 2024) extends distribution matching with mechanisms to incorporate task-specific state-dependent costs. Beyond solvers that primarily target optimal couplings from unpaired marginals, Somnath et al. (2023) and Liu et al. (2023) consider *bridge matching* in settings with *a priori* paired data (e.g., clean/corrupted images or paired biological modalities), where the goal is to exploit the given alignment rather than infer it from scratch via a static SB. Finally, several works extend to multi-marginal settings (Theodoropoulos et al., 2026) or improve computational efficiency of matching-based SB pipelines, for instance via lightweight solvers using Gaussian-mixture parameterizations (Gushchin et al., 2024; Rapakoulias et al., 2024), and via feedback-style formulations (Theodoropoulos et al., 2024).

**Discrete SB with CTMC dynamics.** Recent discrete-space SB solvers also adopt continuous-time Markov chain (CTMC) dynamics, broadly aligned with our control-theoretic perspective. However, unlike our setting, these approaches typically do not cast transport as particle motion constrained by a fixed sparse routing topology, and they do not naturally support application-specific state-dependent running costs for shaping intermediate behavior. Algorithmically, the two works below solve the SB problem via *Iterative Markovian Fitting* (IMF), i.e., the Markov-chain analogue of iterative fitting for SB. For

example, Discrete Diffusion Schrödinger Bridge Matching (DDSBM; (Kim et al., 2024)) targets graph-to-graph translation by generating edit sequences in graph space (transitions correspond to edit operations), rather than edge-feasible routes on a prescribed logistics network. Relatedly, categorical SB methods for generic discrete domains (e.g., token/codebook spaces) similarly rely on discrete-time IMF-style iterative fitting for translation and matching, but are not posed as topology-constrained routing with state-cost shaping (Ksenofontov & Korotin, 2025). Concurrently, Guo et al. (2026) proposed a discrete Schrödinger Bridge formulation based on adjoint matching for CTMC dynamics. While this is closely related to our control-theoretic viewpoint, their formulation focuses on particular choices of the reference transition structure, such as uniform base transition rates, and does not directly address sparse topology-constrained routing with application-specific state-dependent running costs.

**SB for topological signals on graphs.** Topology-aware SB (TSBM; (Yang, 2025)) instead models the evolution of continuous topological signals over graphs and simplicial complexes using Laplacian/Hodge-driven diffusions (admitting closed-form Gaussian bridges in special cases and neural parameterizations more generally). While it follows an SDE-based SB formulation and thus shares the continuous-state SB toolbox, including IPF-style training akin to (De Bortoli et al., 2021), its focus is distinct from edge-by-edge routing semantics and state-cost-controlled particle flows on a fixed sparse transport network. Consequently, its emphasis is on distribution matching for signals defined on a topology, rather than topology-constrained routing of mass through a sparse logistics network.

