# OpenReview forum: "Generalized Schrödinger Bridge on Graphs"
_ICML.cc/2026/Conference — ICML 2026 regular_

### Official Review · Reviewer_cKvu · 2026-03-11

**Soundness:** 4
**Presentation:** 4
**Significance:** 3
**Originality:** 2
**Overall Recommendation:** 5
**Confidence:** 2

**Summary:**

This paper proses GSBoG (Generalized Schrödinger Bridge on Graphs), aiming at solving transportation problem on graph with better scalability by generalizing GSB on continuous domain to discrete graph domain using continuous-time Markov chain. The authors derive a discrete version of the Hopf-Cole transform and introduce a TD loss to handle state-dependent running costs

**Compliance With Llm Reviewing Policy:**

Affirmed.

**Final Justification:**

I will tend to accept this paper. The problem it solved (Generalized Graph SB) is novel and the authors addressed my concern during rebuttal.

**Key Questions For Authors:**

1.  Can the authors provide more detailed analysis on scalability of GSBoG? For example, the training time or memory consumption during experiment, the scaling analysis like runtime or accuracy vs. node numer & edge number. And the largest graph tested has ~10K nodes, from my understanding, it seems a relatively small scale graph.
2. Theorem 3.1 is stated under "mild regularity assumptions,"  will the case $f_t$ depending on $p_t$ complicate strong duality?
3. Using a complex CTMC transportation framework to solve the linear assignment problem is questionable since exact polynomial-time algorithms exist for this task. Can the authors explain the motivation?

**Limitations:**

No limitations are discussed in the main text. I suggest:
1. Discuss the model's ability (or lack thereof) to generalize to unseen graph topologies or different source-target marginals without full retraining.
2. Provide an assessment of the memory and time complexity relative to the number of edges and nodes. While the particle-based approach avoids dense solvers, the per-node potential parameterization still faces memory constraints on truly large-scale graphs

**Strengths And Weaknesses:**

### Strengths
1. **Soundness.** The paper provides a clean and self-consistent derivation of the GSB framework on graphs. Theorem 3.1 establishes the dual representation and optimal control via the potential $V_t$. Prop. 3.4 gives the discrete Hopf-Cole transform. Prop 3.5 and 3.6 derive the  generator-based gIPF and TD objectives. And Algorithm 1 is then proposed to optimize the problem
2. **Presentation.** The paper is well organized. The detailed comparison in Table 1 is particularly effective at making the continuous-to-discrete correspondence explicit.
3. **Significance.** The experiment settings are well-chosen and the resulst are impressive. In supply chain routing task, GSBoG shows compelling result that it implicitly satisfies capacity constraints while maintaining good terminal matching. And the molecular dynamics experiment demonstrates the significant better performance of GSBoG.
4. **Originality.** The running cost mechanism and particle-based formulation are novel for the graph transortation problem. GSBoG's particle-based, trajectory-level formulation is architecturally more scalable than previous method which solve a Hamiltonian BVP over the probability simplex

### Weaknesses
1. While the mapping from continuous to discrete spaces is non-trivial, it remains a relatively direct extension of the DeepGSB framework (Liu et al., 2022b). The core algorithms like the IPF update and the TD regularization are directly inherited from DeepGSB.  The paper would benefit from a more explicit discussion of what challenges or surprises arise specifically from the discrete/graph structure that do not have counterparts in the continuous case.
2. Though the supply chain experiment shows that GSBoG can handle complex tasks where GrSB will OOM, a more comprehensive evaluation of scalability should be conducted to claim scalability as a key advantage.
3. The empirical evaluation relies heavily on comparing GSBoG against naive baselines (Attraction-Flow) or methods that fail to scale (GrSB). GrSB's failure due to OOM errors provides a "low bar" for scalability. The authors do not compare against recent neural graph transport or discrete-diffusion-based bridge matching methods, which could have been adapted for a more competitive evaluation.

---

> ### Author Rebuttal · Authors · 2026-03-31
>
> We thank the reviewer for their positive comments and constructive criticism. Below, we address the raised points.
> ## **1. Contribution and novelty**
> While GSBoG shares the high-level goal of generalized Schrödinger Bridge (SB) transport with DeepGSB, it is not a discrete extension. DeepGSB is formulated in continuous space, whereas in GSBoG, mass moves along feasible edges on a graph under a controlled CTMC. Reformulating the training framework on the graph is non-trivial, as the dynamics follow the graph continuity equation and the objective is a KL-regularized path-space problem over CTMC measures, for which we can not use continuous stochastic calculus tools (e.g., Ito’s formula). Instead, we rely on the CTMC infinitesimal generator and Dynkin's formula to yield discrete potential-generator identities and graph-native objectives. Moreover, the TD term is not borrowed heuristically from DeepGSB; it arises to enforce local consistency after showing that running-cost terms cancel inside the IPF objective. Therefore, GSBoG makes substantial new theoretical contributions by establishing the discrete control and training framework for generalized SB transport on sparse graphs.
> ## **2. Conditions for Theorem 3.1**
> Theorem 3.1 should be read as the coupled primal-dual optimality system for an optimal solution. When $f_t$ depends on  $p_t$, a strong-duality claim requires lower boundedness, continuity, and convexity on the induced marginal cost. These assumptions are standard in the variational mean-field-control and distributional cost-aware steering literature. In our experiments, the running costs are either convex with respect to the marginal (supply chain) or are only state-dependent (assignment, folding).
> ## **3. Comparison with Discrete Neural Methods**
> Recent discrete SB-based methods are not drop-in baselines, as they differ fundamentally from our setting. TSBM is conceptually the closest to GSBoG, due to its topology-aware formulation, but it models continuous signals with topology-aware diffusion operators. CSBM is a bridge matching method designed for categorical translation on unstructured discrete spaces, while DDSBM is also a bridge matching framework for graph-to-graph transformation through node/edge edits in graph space. Thus, while these methods share high-level traits with our GSBoG, they can not learn an executable CTMC policy to solve the same finite-horizon, endpoint-constrained graph transport problem. For completeness, we forcibly adapted TSBM to our supply-chain and protein-folding tasks, and its weaker empirical performance in terms of terminal mismatch reflects this. In contrast, CSBM and DDSBM can not be meaningfully adapted to our setting, as they lack the capacity to handle non-trivial reference transition rates.
> ||Supply Chain|Protein Folding|
> |-|-|-|
> |TSBM|0.55|0.96|
> |GSBoG|**0.03**|**0.01**|
>
> Moreover, generic discrete diffusion and discrete flow-matching methods are not well-suited to our setting, as they assume a predefined, simple forward corruption process (e.g., masking or uniform noise) and learn only its reversal. This paradigm suits generic discrete generation, but not for transport on a fixed graph with general reference CTMC dynamics and arbitrary prescribed marginals.
>
> We also stress that the paper's baselines are not weak heuristics, but principled OT-based baselines tailored for graph-transport problems. The evaluation tests whether a method can transport mass accurately while achieving good deployment behavior. From that perspective, the results show that existing methods used in this setting are either computationally fragile at this scale or substantially less effective at controlling intermediate behavior.
> ## **4. Scalability**
> For a detailed computational analysis, we refer the reviewer to the 'Scalability' paragraph in our response to Reviewer 28Ro. Additionally, we constructed synthetic sparse graphs with 100K and 1M nodes, with source and target distributions each supported on 10 nodes at 0.1% connectivity. The low terminal mismatch shows that GSBoG still transports mass accurately at a very large scale, beyond the capacity of existing methods.
> |Nodes|Framework|Total Variation|Time (hours:min)|Memory (GB)|
> |-|-|-|-|-|
> |100k|Dynamic OT|-|-|-|
> |100k|Graph SB|-|-|-|
> |100k|GSBoG|0.03|1:07|2.9|
> |1 M|GSBoG|0.06|6:32|26|
> ## **5. Assignment problem**
> We agree that, for the static linear assignment problem, exact polynomial-time algorithms are available, and GSBoG is not intended to compete with them. The assignment example tests whether a learned CTMC policy can reproduce the correct coupling when pairwise costs are encoded as state costs on a graph — a controlled test of the framework’s expressivity where the ground truth is known exactly. We will revise the manuscript to better clarify this motivation.
> ## **6. Limitations**
> We thank the reviewer for this suggestion. In the revision, we will add a section discussing the limitations of our current framework.

---

> > ### Author Rebuttal · Reviewer_cKvu · 2026-04-02
> >
> > Thanks for the reponses. I will raise my score to 5.

---

> > > ### Author Response · Authors · 2026-04-06
> > >
> > > We sincerely thank the reviewer for their comment and appreciate their acknowledgment that our responses have effectively addressed all their concerns.

---

### Official Review · Reviewer_28Ro · 2026-03-12

**Soundness:** 3
**Presentation:** 3
**Significance:** 3
**Originality:** 3
**Overall Recommendation:** 5
**Confidence:** 3

**Summary:**

The paper presents Generalized Schrödinger Bridge on Graphs (GSBOG) which is a framework for optimal transport on discrete graph structures using continuous-time Markov chain (CTMC) dynamics. Unlike previous graph-based Schrödinger Bridge methods that rely on global solvers, GSBOG uses a particle-based, data-driven approach. The method combines a generalized Iterative Proportional Fitting (gIPF) objective to match endpoint marginals with a temporal-difference (TD) objective to enforce consistency with state-dependent running costs. The authors demonstrate the effectiveness of GSBOG across supply chain routing, assignment tasks, and molecular dynamics folding, showing improved scalability and cost-awareness compared to baselines like GrSB and Attraction Flow.

**Compliance With Llm Reviewing Policy:**

Affirmed.

**Final Justification:**

The rebuttal partially addressed my main concerns, but I'm skeptical about the practicality of the proposed framework. I thus stick to the conservative rating of this paper.

---

After the second rebuttal from the authors, I no longer have concerns since the authors provided stronger experimental results.

**Key Questions For Authors:**

1. Regarding Equation 19, how do you ensure the local ratios of the potentials remain numerically stable during the early phases of training before the gIPF has converged?

2. In the supply chain experiment, you mentioned GrSB failed due to memory exhaustion; what is the specific O-complexity of GSBOG relative to the number of edges and particles?

3. Have you considered using a more structured architecture than an MLP for the node embeddings, such as a Graph Neural Network (GNN), to better capture the local topology in the potentials?

**Limitations:**

yes

**Strengths And Weaknesses:**

Strengths:

1. The paper provides the first data-driven formulation of generalized Schrödinger bridges specifically for arbitrary graph topologies, which is a significant practical contribution.

2. The method effectively addresses the scalability limitations of existing graph optimal transport methods by avoiding dense global solvers and using a local, particle-based rollout strategy.

3. The inclusion of a temporal-difference (TD) objective is a clever way to inject running costs into the learning process, which is often a challenge in entropic transport formulations.

4. The experiments are diverse and compelling, particularly the chignolin folding task, which demonstrates the ability to steer rare-event kinetics in complex spaces.

---

Weaknesses:

1. The performance is sensitive to the tradeoff parameter $\lambda_{TD}$, as shown in the ablation study, where increasing accuracy in cost-consistency leads to a noticeable drop in terminal distribution matching.

2. While the method is more scalable than global solvers, the reliance on iterative rollouts and sampling could still be computationally demanding for extremely large graphs or long horizons.

3. The choice of MLP parameterization for the log-potentials is standard, but more discussion on how the node embeddings handle very sparse or disjoint graph components would have been beneficial.

---

> ### Author Rebuttal · Authors · 2026-03-31
>
> We thank the reviewer for their positive comments and constructive criticism. In the following, we address the points raised by the reviewer.
> ## **1. Scalability**
> We agree that GSBoG still incurs rollout cost, particularly for very long horizons. However, GSBoG is substantially more scalable than global graph transport solvers because it avoids solving a global optimization problem over all nodes and time indices. Instead, it learns trajectory-level CTMC policies, based on sampling particle trajectories around local high probability regions with the optimal controlled rate recovered as $u_t^\star(y,x)=r_t(y,x)\frac{φ_t(y)}{φ_t(x)}$  from local ratios along edges $(x,y)$. As a result, each rollout step depends on the local degree $\deg(x)$, instead of the total number of nodes $n$. Hence, the cost of GSBoG scales — at worst — according to the local branching factor $\Delta=\max_x \deg(x)$, which is the main scalability advantage of our framework, especially on sparse graphs, where $\Delta\ll n$.  This distinction becomes dramatic in the supply chain setup, since $n=9559$, but the maximum degree is 6, so each particle step ever interacts with at most six neighbors rather than all 9559 nodes. Consequently,  for $M$ outer iterations, $H$ time steps, and batch size $B$, the cost is $O(MBHΔ)$, with memory $O(n\Delta+B\Delta)$.
>
> By contrast, dynamic OT on graphs discretizes a global flux optimization over all nodes and edges across time, with sparse time-expanded memory $O(H(m+n))$, and time on the order of $O(ΔH^2n^2)$, where  $m$ is the total number of edges. Graph SB [1] formulates SB on graphs as a two-point boundary value problem on the graph probability simplex across all nodes and time steps, giving $O((Hn)^3)$ complexity, and $O((Hn)^2)$ memory. This explains why global solvers scale poorly, since their complexity grows with the total number of nodes and time steps; whereas GSBoG scales with the local branching factor per visited node, which becomes crucial on sparse graphs. Lastly, we compare the empirical training time and memory needed across methods.
> ||Solving Time (hours: min)|Memory (MB)|
> |-|-|-|
> |Dynamic OT|2:38|7,109|
> |Graph SB [1]|NA|NA|
> |GSBoG|0:55|550|
>
> ## **2. Node Embedding and GNN**
> In our implementation, the temporal trunk first produces a shared time feature $h(t)\in\mathbb{R}^d$, and each node $x$ is associated with a trainable embedding $e_x$ and bias $b_x$. The learned log-potential has the form $Y_t(x)=\langle e_x,h(t)\rangle+b_x$. For $(t,x)$ the model learns a time-dependent node potential, and the dynamics at $x$ depend only on $x$ and its neighbors.
>
> We chose the current MLP-style parameterization to isolate the contribution of the GSBoG framework and preserve scalability. Crucially, the network does not need to materialize the full graph, as it evaluates $Y_t(x)$ locally per node, with topology being enforced by the sparse reference generator, avoiding storing dense $N \times N$ matrices. By contrast, GNN-based potentials, while expressive, would introduce graph-wide message passing at every time step, increasing per-iteration cost and memory. That said, we agree that using an explicitly graph-structured representation (e.g., GNN-based potentials) is a natural extension, as it could improve information sharing. We will clarify in the revision the motivation behind our current architecture, and explicitly list GNN architectures as a promising future direction.
> ## **3. Stability of local ratios in Eq. 19**
> Regarding Eq. 19, we do not compute local ratios by explicitly dividing two estimated potentials. Instead, we parameterize $Y_t(x)=\log φ_t(x)$ and $\hat Y_t(x)=\log \hat φ_t(x)$, so the local ratio in Eq. (19) is represented as $\frac{φ_t(y)}{φ_t(x)}=\exp\big(Y_t(y)-Y_t(x)\big)=\exp(Z_t(y,x)).$ This removes division-related singularities, although the exponential alone does not guarantee stability. In practice, the MLP initialization makes the logits start near zero, so the controlled rates begin close to the reference CTMC rather than at extreme values. At each simulation step, the transition probabilities are normalized, so the rollout always remains a valid probability distribution, such that the rollout scheme provides sufficiently broad early coverage of the graph, and contributes to the stability of the gradient estimates. Moreover, numerical stability in the early phases does not rely on gIPF having already converged. The TD term enforces local generator consistency by requiring the one-step increments of the learned log-potentials to match the generator-predicted increments. This directly regularizes the edge-wise logits $Z_t(y,x)$ and $\hat Z_t(y,x)$, preventing erratic local-ratio updates before endpoint matching has fully stabilized.
> ## **4. Ablation Study on $λ_{TD}$**
> For an ablation study on $λ_{TD}$, we refer the reviewer to the 'Task-specific cost effect' paragraph in our response to Reviewer Nxbg.
>
> ---
> References
> 1. Dynamical Schrödinger Bridge Problems on Graphs

---

> > ### Author Rebuttal · Reviewer_28Ro · 2026-04-02
> >
> > Thank the author for the detailed rebuttal. My main concerns are partially resolved, particularly around numerical stability, representation choice, and the lack of a more systematic scaling study.
> >
> > 1. Your answer on Eq. (19) mainly explains how explicit division is avoided, but not why the exponential edge logits remain stable early in training. Can you provide either a formal theoretical bound, or an empirical result showing that $Z_t(y, x)$ does not become pathological before convergence?
> >
> > 2. The scalability argument is reasonable, but could you provide a true scaling study over graph size, edge count, horizon, and rollout budget, rather than only complexity discussion plus a few runtime snapshots? That would make the scalability claim much more convincing.
> >
> > 3. You justify the current MLP-style node embedding on efficiency grounds, but this does not yet show whether the parameterization is sufficient on highly sparse, weakly connected, or heterogeneous graphs. Do you believe the current architecture is adequate there, or is a graph-structured encoder likely necessary?
> >
> > ---
> >
> > After the second rebuttal, the authors cleared all my doubts and concerns. Therefore, I will raise my score from 4 to 5. I believe this is a solid work.

---

> > > ### Author Response · Authors · 2026-04-06
> > >
> > > We sincerely thank the reviewer for their comment. Below, we address the final concerns:
> > > ## **1. Stability**
> > > We address this point empirically by the learned control logits that enter the exponential rates in Eq. (19), namely (i.e., $Z_t$ for the forward, $\hat Z_t$ for the backward pass).
> > > Across the supply chain and protein-folding experiments, these logits exhibit a brief rise following initialization, and then stabilize and converge, and importantly, do not exhibit any runaway growth, as shown in the attached figure on: https://zenodo.org/records/19435494.
> > > Over the full training run, the maximum observed control logit was 3.05 in the supply-chain setting and 4.8 in the folding setting. Thus, while Eq. (19) uses an exponential parameterization, we empirically verify the numerical stability of our framework through training. We will include this figure and summary in the revision.
> > > ## **2. Scalability**
> > > We present explicit ablations to demonstrate the scalability of GSBoG. Concretely, we evaluate GSBoG for varying graph size, edge count, horizon, and rollout budget.
> > >
> > > On synthetic graphs at fixed sparsity, GSBoG scales favorably as nodes increase from 50k to 1M nodes while maintaining low terminal TV. On a 10k-node graph, increasing density from 0.5% to 5% keeps low terminal mismatch, with memory increasing from 7.9 GB to 20.1 GB.
> > > ||Dynamic OT|||GSBoG|||
> > > |-|-|-|-|-|-|-|
> > > |Num. of Nodes |TV|Memory (GB)|Time(hours: sec)|TV|Memory (GB)|Time(hours: sec)|
> > > |50K|0.02|22.0|4:39|0.03|1.4|1:02|
> > > |100K|-|-|-|0.03|2.6|2:08|
> > > |1M|-|-|-|0.06|26.9|6:32|
> > > |Edges (density %)|
> > > |500K (0.5%)|0.03|12.5|4:15|0.02|7.9|2:50|
> > > |1M (1%)|0.02|23.8|5:58|0.03|15.5|3:46|
> > > |5M (5%)|-|-|-|0.03|20.1|4:22|
> > >
> > >
> > > On the supply-chain graph, we increase the number of steps up to 400 steps, which preserves TV at about 0.03; lastly, increasing the rollout budget from 1k to 4k improves TV from 0.11 to 0.02, due to enhanced exploration. By contrast, it is underlined that the Dynamic OT baseline becomes memory- or time-prohibitive much earlier and does not complete in several larger regimes. We will add the full tables and plots in the revision.
> > >
> > > ||Dynamic OT|||GSBoG|||
> > > |-|-|-|-|-|-|-|
> > > |Steps|TV|Memory (GB)|Time (hours: sec)|TV|Memory (GB)|Time (hours: sec)|
> > > |100|0.03|7.0|2:39|0.03|0.51|0:55|
> > > |200|0.01|18.1|7:37|0.03|0.66|2:15|
> > > |400|-|-|-|0.03|0.79|3:31|
> > > |Rollout Budget|
> > > |1K|0.03|6.9|2:37|0.11|0.23|0:34|
> > > |2K|0.03|7.0|2:39|0.03|0.53|0:55|
> > > |4K|0.02|7.3|2:50|0.02|0.71|1:58|
> > >
> > > ## **3. Parameterization**
> > > We believe the current architecture is the better option for the studied graph regime, not only because of its efficiency, but also due to its alignment with the GSBoG control law. In our implementation, the network approximates the scalar log-potential $Y_t(x)$, where $x$ denotes the current node at time $t$. Importantly, this quantity depends only on the current state $x$ and time $t$, thus an MLP-style parametrization for estimating $Y_t(x)$ is sufficient and more computationally efficient, rather than incorporating the full graph structure through a GNN.
> > >
> > > Explicitly, our loss in Eq. 26-27 requires computing $Z_t(y,x) = Y_t(y) - Y_t(x), \ y \in N(x),$ for neighboring nodes $y$ of $x$. This motivates the current design, as the loss only depends on pairwise potential differences between adjacent nodes. Concretely, we use
> > >
> > > $$e_x = \mathrm{NN}_1(x), \quad e_y = \mathrm{NN}_1(y), \quad e_t = \mathrm{NN}_2(t) \in \mathbb{R}^d,$$
> > > where $y \in N(x)$. Since all neighboring embeddings $e_y$ for $y \in N(x)$ can be obtained together with $e_x$ through a single pass of $\mathrm{NN}_1$ over $\{x\} \cup N(x)$, the resulting implementation is simple and efficient. In particular, for a queried node $x$ and its neighborhood $N(x)$, we obtain the node embeddings
> > >
> > > $$e_{x\cup N(x)} \in \mathbb{R}^{(1+|N(x)|)\times d}$$
> > > in a single pass, and then combine them with a shared temporal feature $e_t\in\mathbb{R}^d$ through
> > > $$Y_t(z)=\langle e_z,e_t\rangle+b_z, \qquad z\in \{x\}\cup N(x),$$
> > > so that $Y_t(x)$ and $\{Y_t(y)\}_{y\in N(x)}$ are computed together in one pass.
> > > Thus, since the control depends on nodewise scalar potentials and local neighbor differences, rather than graph-wide message passing, the benefit of GNNs is not clear in our setting.
> > >
> > > Lastly, our experiments show that the current parameterization remains effective in extremely sparse, bottlenecked regimes. In the supply-chain experiment, the graph has $N=9559$ and 99.97% sparsity,  with very few feasible routes; similar bottlenecks also arise in the protein-folding MSM. Yet GSBoG still achieves near-optimal terminal matching while optimizing the intermediate trajectory, indicating that the parameterization already captures the relevant topology for large-scale sparse transport networks. Heterogeneous graphs are an interesting extension, e.g., multiple product types in supply chains, which can be effectively handled through type-specific embeddings or parallel potential networks with coupling constraints at shared nodes.

---

### Official Review · Reviewer_Nxbg · 2026-03-12

**Soundness:** 4
**Presentation:** 4
**Significance:** 2
**Originality:** 2
**Overall Recommendation:** 4
**Confidence:** 2

**Summary:**

This paper considers a generalized version of the Schrödinger bridge problem where the start and end distrbutions are supported on graphs. Using a reference process on that graph and task-specific regularizers the authors find better intermediate states and distribution matching compared to other methods.

**Compliance With Llm Reviewing Policy:**

Affirmed.

**Key Questions For Authors:**

1.	Clarification of novelty with respect to [1].
Can the authors clearly articulate, both conceptually and mathematically, how their formulation differs from Topological Schrödinger Bridge Matching [1]? In particular, under which assumptions (if any) does Eq. (13) reduce to the TSBP formulation of [1], and what aspects of the current framework strictly go beyond it? A precise comparison would significantly clarify the paper’s originality.
2.	Role and impact of the additional regularizers.
How critical are the task specific regularizers in Eq. (13) for the reported improvements? An ablation study (or explicit discussion) isolating the effect of these regularizers would help assess whether they are the main source of improvement over existing graph based SB approaches.
3.	Comparison against [1] in experiments.
Is it possible to include [1] as a baseline in at least one of the experimental settings, or to explain in detail why such a comparison would not be meaningful or feasible? This comparison would help determine whether the proposed generalization yields empirical benefits beyond existing topological SB methods.
4.	Scope of applicability beyond fixed graphs.
The paper emphasizes a fixed graph topology. How would the approach extend (or fail to extend) to settings with evolving graphs, uncertain topology, or partially observed connectivity? Clarifying this would help delimit the practical scope of the contribution.

**Limitations:**

The limitations of the approach are only partially discussed. In particular, the paper would benefit from a more explicit discussion of (i) how strongly the method depends on the assumption of a fixed and known graph topology, (ii) the sensitivity of the results to the choice of reference process and task specific regularizers, and (iii) the extent to which the proposed framework provides guarantees versus empirical improvements for intermediate state behavior. Additionally, since the method can be applied to real world transport and allocation problems, a brief discussion of potential misuse or unintended consequences would strengthen the broader impact section.

**Strengths And Weaknesses:**

### Strengths:
- The experiments on real world data (supply-demand network and molecule folding) are interesting and showcase the relevance of the studied problem. Importantly, the experiments do not consist of simple benchmarking but application-relevant tasks carefully designed for this paper. Moreover the very detailed description of these experiments in the appendix adds value.
- The math is precise and the results clearly stated
- The paper is well written, and the tables, explanatory figures (1 and 2) and algorithm 1 help the reader to understand the main points



### Weaknesses/questions:
#### major:
- I am not sure about how this paper relates to the work [1] (which the authors mention in the appendix). In my current understanding, the paper [1] introduces the topological Schrödinger Bridge problem, where the goal is to transport one signal on e.g. a graph to another where the intermediate states are sensible. This is however also one of the main contributions of the current paper, limitting its originality and significance. As I see it, the new contribution here is the freedom to put additional regularizations on the obtained paths (in the form of $\mathbb{E}_{p^u}\mathbb{E}_t(f_t(X_t,p_t))$ in eq. 13). The first part of eq.13 is equivalent to to the "TSBP" equation from [1] in the beginning of their section 3 with a slightly different reference process. I welcome the authors to clarify the relation between their work and [1] and any misunderstanding I might have.
- in the context of the above point I find the following statements problematic:
    - "In contrast, our setting assumes a fixed graph topology and studies the problem of transporting probability mass over the given node space X . To the best of our knowledge, this perspective has not been systematically investigated in the machine learning literature." (l.193 left)
    - "Closest to our setting is the dynamical SB on graphs (Chow et al., 2022)" (l.85 left)
- in the experiments the authors should compare against the method from [1] to study whether their approach outperforms the existing one, i.e., whether the additional regularizers in eq.13 have a significant effect

#### minor:
- The appendices should be more frequently referred to in the main text as they add value and the reader should be aware of them after reading the main text. From my reading, I didn't find any reference to appendix A (proofs) and smilar for appendix D (related work).
- I suggest to change figure one to also show a later time point where the mass is successfully transported to the target, as currently it looks like the process ends in an intermediate state

[1] Yang, Maosheng. "Topological Schrödinger Bridge Matching." arXiv preprint arXiv:2504.04799 (2025).

Overall I find this to be a strong paper where the unclear relation to [1] is my only major concern.

---

> ### Author Rebuttal · Authors · 2026-03-31
>
> We thank the reviewer for the encouraging comments and useful remarks. Below, we address the raised points.
> ## **1. Relation to TSBM**
> We agree that the relation to Topological Schrödinger Bridge Matching (TSBM) should be better illustrated. Both methods are Schrödinger-bridge-based and topology-aware, but they address different objects and encode topology differently. The main conceptual distinction is transported and how topology is enforced.
>
> TSBM is a continuous-state, SDE-based framework for matching continuous topological signals, with a linear-SDE reference process. Its topology awareness is operator-based, entering through Laplacian/Hodge-type diffusion operators that shape the dynamics. By contrast, GSBoG is discrete-state and CTMC-based, where particles move on the graph itself. Here, topology is enforced as a hard constraint; transitions are allowed only along existing edges and are exactly zero otherwise, which is fundamentally different from TSBM’s operator-based notion of topology enforcement.
>
> Thus, despite addressing seemingly similar objectives, Eq. (13) does not reduce to TSBP, even when the running cost $f=0$, as the two problems remain structurally different. GSBoG optimizes CTMC path measures on a finite graph, whereas TSBP optimizes over diffusion path measures on \mathbb{R}^d. Reconciling the two would require fundamentally changing the state space, the reference dynamics, and the meaning of the transported object.
>
> ||TSBM|GSBoG|
> |-|-|-|
> |State Space |$\mathbb R^d$|$\mathcal X$|
> |Reference Process|Cont. SDE|CTMC|
> |Matching|Continuous signals over nodes|Particles on the nodes|
>
> This distinction becomes particularly clear in our experiments. To compare in the supply-chain and protein-folding settings, we had to forcibly adapt our setup, as TSBM requires lifting each particle at a graph node to an N-dimensional one-hot signal and evolving it in a continuous signal space using the graph only through a continuous interpolant path. The resulting trajectory is still not a discrete CTMC particle path, but a continuous node-signal that must be projected back to node marginals at evaluation. More importantly, this breaks the native “particle-at-a-node” semantics, making terminal evaluation especially brittle for our sparse boundary marginals. As a result, the forced adaptation yields very poor performance, which we interpret as evidence of a modeling mismatch rather than a negative statement about TSBM itself.
> |Total Variation|Supply Chain |Protein Folding|
> |-|-|-|
> |TSBM|0.55|0.96|
> |GSBoG|**0.03**|**0.01**|
>
> Consequently, our contribution goes beyond incorporating task-specific state costs or transferring an existing SB framework to graph data. GSBoG is entirely graph-native: starting from CTMC dynamics and the discrete continuity equation on the graph, we derive the optimality conditions (optimal policy and Hopf–Cole equations) and a scalable training objective (gIPF+TD), all defined directly on the graph topology. To our knowledge, this makes GSBoG the first data-driven generalized Schrödinger bridge framework for executable, cost-aware transport on arbitrary graphs.
> ## **2. Limitations**
> In the revision, we will explicitly discuss the limitations of our framework. The framework is currently scoped only to known fixed graph topologies, as the CTMC reference generator is defined on a graph $G=(\mathcal X,\mathcal E)$, the controlled rates are constrained to the same support. We note that time-varying graphs should be approachable by introducing a time-dependent support $\mathcal E_t$ with a time-varying reference generator. Contrarily, handling partially observed connectivity with the current framework is a non-trivial extension, as rollout feasibility, and the loss evaluation all require knowing the admissible transitions; we view this as an important open direction.
> ## **3. Task-specific cost effect**
> Including task-specific costs in Eq. 13 greatly improves intermediate trajectory quality. Empirically, the existing $f=0$ vs $f\neq 0$ comparisons verify this effect; in the supply chain, peak occupancy was halved while preserving accurate endpoint matching.  The parameter $λ_{TD}$ balances the IPF and TD gradient signals so that neither dominates training, improving stability and convergence.
>
> Our ablation on $λ_{TD}$ demonstrates a clear knee effect; with the main benefit being achieved even with small positive $λ_{TD}$, as congestion drops substantially and terminal mismatch remains virtually unchanged. Beyond that point, larger $λ_{TD}$ yields only marginal further gains, while the method remains robust overall and terminal matching changes only modestly at high values.
> |$λ_{TD}$|TV|Peak occupancy|
> |-|-|-|
> |0 | 0.02 | 320|
> |0.1 | 0.03 | 274|
> |0.5 | 0.04 |240|
> |1|0.05|237|
>
> ## **4. Minor clarifications.**
> In the revision, we will add pointers to the appendix and revise Fig. 1 to make the successful arrival at the target visually explicit.

---

> > ### Author Rebuttal · Reviewer_Nxbg · 2026-04-03
> >
> > I find this a strong rebuttal. The authors addressed my point. I would raise to score from 4 to 6., and significance from 2 to 3.

---

> > > ### Author Response · Authors · 2026-04-06
> > >
> > > We sincerely thank the reviewer for their comment and appreciate their acknowledgment that our responses have effectively addressed their concerns.

---

### Decision · Program_Chairs · 2026-04-30

**Decision:**

Accept (regular)

**Comment:**

The paper considers the challenging problem of building schrodinger bridges on graphs . The trajectory level policy respecting the graph bsed costs and source/target constraints is learned. A TD objective is also introduced for enforcing the graph based consistency. The algo can be understood as alternating between source/target constraints and the consistency ones. Simulations show the efficacy of the method.

All the reviews were positive and most concerns of reviewers were answered in the author rebuttal.

Based on this, I recommend an accept.